# GenCoGS: Generative Completion-based 3D Gaussian Splatting for High-Fidelity Few-Shot Novel View Synthesis

## Abstract

Conventional few-shot novel view synthesis (NVS) methods using 3D Gaussian Splatting (3DGS) have demonstrated significance, but remain constrained by their overdependence on the limited information from training views. Their unsatisfactory scene completion capability would underrepresent those scene regions either unobserved in training views or with local details and thus cause floating artifacts against high fidelity. To address these challenges, we propose GenCoGS, a unified 3DGS-based few-shot NVS method focusing on initializing and optimizing 3DGS representation using generative completion-based strategies to enhance scene completion. Specifically, our generative point cloud completion-based strategy produces and filters complementary points toward a complete point cloud with refined structural and appearance information for Gaussian initialization; The generative pseudo view completion-based strategy leverages an image-to-video diffusion model to synthesize complete pseudo views, which benefits Gaussian optimization especially within unobserved scene regions and mitigates hallucination for less appearance distortion. Integrating both strategies enables accurate and coherent scene completion for high-fidelity few-shot NVS. Extensive experiments on three benchmark datasets demonstrate the superiority of our GenCoGS for few-shot NVS evaluated in common metrics compared to baseline methods. Compared to those 3DGS-based few-shot NVS methods, our GenCoGS achieves improvements of up to 2.40 dB, 0.08 and 0.125 in PSNR, SSIM and LPIPS.

## 1 Introduction

Few-shot novel view synthesis (NVS) aims to synthesize images of the target scene from unseen viewpoints given a set of sparse images from limited known viewpoints. This task demonstrates significant practical value in high-quality rendering upon data sparsity (Zhu et al., 2024). Existing methods focus on adapting general NVS models, *e.g.*, Neural Radiance Fields (NeRF) (Mildenhall et al., 2020) and 3D Gaussian Splatting (3DGS) (Kerbl et al., 2023), for few-shot NVS via prior knowledge (Chen et al., 2021; Niemeyer et al., 2022; Kulhánek et al., 2022; Yu et al., 2021a; Wang et al., 2023; Yang et al., 2023; Li et al., 2024a; Zhu et al., 2024; Paliwal et al., 2024; Zhang et al., 2024).

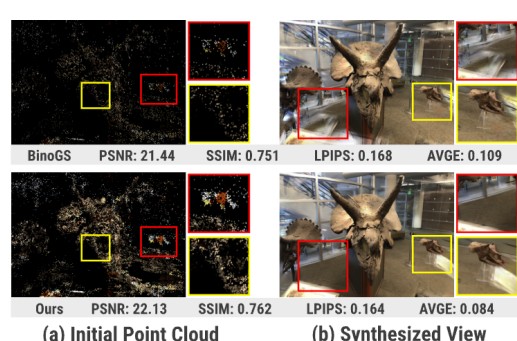

Figure 1: Limited scene completion capability of existing 3DGS-based few-shot methods, represented by (a) insufficient local details due to the incomplete initialization of Gaussians; and (b) floating artifacts in unobserved regions due to the optimization guided by pseudo views.

In particular, those high-fidelity and efficient few-shot NVS methods based on 3DGS are generally characterized by a two-phase pipeline: (1) 3D Gaussian initialization based on fused stereo points generated from training views (Zhu et al., 2024) or image pixels in training views using corresponding depth maps (Paliwal et al., 2024), and (2) 3D Gaussian optimization based on enhanced priors from training views

with additional supervision from sampled pseudo views (Zhu et al., 2024; Zhang et al., 2024). Despite the significant performances achieved, these methods are fundamentally confined by the nature of solely leveraging observed information, causing considerably less competent results within certain scene regions. As shown in Figure 1 (a), the initial Gaussians unsatisfactorily represent the scene's structure and appearance in those regions unobserved in training views or with local details; Meanwhile, pseudo views sampled from training views contribute primarily to the observed regions during Gaussian optimization, but lead to floating artifacts within the unobserved regions, as illustrated in Figure 1 (b). These challenges suggest that these methods lack the human *imagination* for imagery generation as scene completion (Pearson, 2019). It inspires us to explore if few-shot NVS, which is less-constrained and under-determined, can be transformed into a sufficiently constrained and observed task by exploiting the mechanism of human imagination.

Considering the notable completion capabilities of recently boosted generative models (Song et al., 2020; Yu et al., 2024a; Wu et al., 2024), we propose a novel unified few-shot NVS method, **Gen**erative **Co**mpletion-based 3D**GS** (**GenCoGS**), to address the aforementioned challenges. This unified method is characterized by two *generative completion-based strategies* on initializing and optimizing scene representation for 3DGS. The former strategy generates a complementary point set and filters this point set to complete the initial point cloud obtained by the SfM (Zhu et al., 2024) regarding structural and appearance details for 3D Gaussian initialization. The latter strategy for 3D Gaussian optimization adopts a perturbed camera trajectory to sample pseudo camera poses probably covering unobserved regions, and an image-to-video (I2V) diffusion model (Yu et al., 2024a) for conditional completion of pseudo views; Meanwhile, a generative consistency loss is designed to provide additional supervision. Both strategies jointly enhance the 3DGS' capability of scene completion while mitigating appearance distortion and floating artifacts caused by the hallucination of generative models (Aithal et al., 2024). Extensive experiments on LLFF (Mildenhall et al., 2019), DTU (Jensen et al., 2014) and Shiny (Wizadwongsa et al., 2021) benchmark datasets, demonstrate that GenCoGS can achieve the state-of-the-art performance under representative few-shot settings with 3, 6 and 9 input training views. The contributions of this paper can be summarized as follows:

- Inspired by the mechanism of human imagination, we propose a unified few-shot NVS method based on generative completion with focus on initializing and optimizing scene representation.

- To the best of our knowledge, we design, for the first time, a generative point cloud completion-based Gaussian initialization strategy leveraging complementary point generation and filtering; and a generative pseudo view completion-based Gaussian optimization strategy exploiting image-to-video diffusion models against hallucination.

- Based on the scene completion capability, the proposed method can outperform representative few-shot NVS solutions across three benchmark datasets.

## 2 RELATED WORKS

**Few-shot Novel View Synthesis**  Few-shot NVS aims to reconstruct accurate and visually compelling 3D scenes from sparse training views, yet suffers from geometric–radiance ambiguity due to insufficient observations. NeRF-based methods mitigate overfitting through strategies such as geometric and color regularization (Niemeyer et al., 2022), depth supervision (Deng et al., 2022), depth distillation (Wang et al., 2023), and generalizable priors via pretrained models (Yu et al., 2021a; Chen et al., 2021; Li et al., 2024b). Despite these advances, implicit MLP-based representations remain computationally demanding and challenging to combine with explicit 3D scene models. Explicit 3DGS-based methods offer advantages in rendering efficiency and quality and have introduced dedicated regularizations to handle sparse inputs. Notably, FSGS (Zhu et al., 2024) and DNGaussian (Li et al., 2024a) use sparse depth supervision to align Gaussians with geometric priors, while CoherentGS (Paliwal et al., 2024) ensures spatial coherence through optical flow constraints.

Nevertheless, these methods are constrained to the observed regions in training views and struggle to model unobserved structure. Unlike prior-based methods, our GenCoGS performs generative completion over unobserved regions by employing strategies on Gaussian initialization and optimization, which jointly enable high-fidelity few-shot NVS with structurally sound and realistic results.

**Diffusion Priors for Novel View Synthesis**  Recent advances in diffusion models have suggested their utility as informative priors for text-driven 3D generation. DreamFusion (Poole et al., 2022)

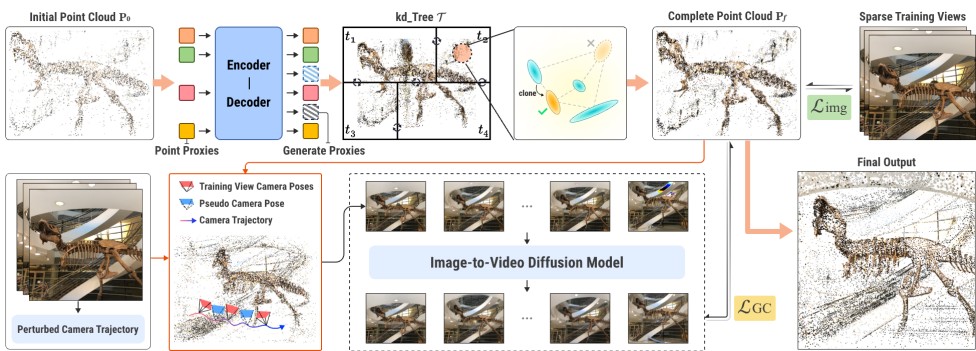

Figure 2: Pipeline of the proposed GenCoGS unified by two generative completion-based strategies on Gaussian initialization and optimization, *i.e.*, GCGI with complementary points and GCGO with pseudo views, for high-fidelity few-shot NVS.

adopts score distillation sampling to leverage pre-trained 2D diffusion models for 3D object synthesis from text prompts, influencing subsequent studies (Tang et al., 2023a; Yi et al., 2024). To improve 3D consistency, Zero-1-to-3 (Liu et al., 2023) and MVDream (Shi et al., 2023) incorporate 3D-aware learning into diffusion models, though they depend on large-scale training data and computation-expensive pipelines. Alternative methods, such as HiFi-123 (Yu et al., 2024b) and Make-It-3D (Tang et al., 2023b), employ a single image with diffusion-based priors for 3D reconstruction but require per-scene optimization that limits scalability. The successes of these methods in 3D generation or reconstruction, however, have exhibited limitations in high-fidelity few-shot NVS.

Meanwhile, ReconFusion (Wu et al., 2024) and IPSM (Wang et al., 2024) demonstrate that diffusion-guided NeRF and 3DGS can accomplish high-quality few-shot NVS using 2D diffusion-based priors. To ensure multi-view consistency, image-to-video diffusion models have been adapted with camera-controlled generation techniques (Blattmann et al., 2023; Chen et al., 2024; Melas-Kyriazi et al., 2024). ViewCrafter (Yu et al., 2024a), CAT3D (Gao et al., 2024) and ReconX (Liu et al., 2025) have further extended this approach to the few-shot setting by integrating image-to-video diffusion models with iterative point cloud refinement. However, these attempts tend to hallucinate within the target scene's unobserved regions, causing structural and appearance inconsistencies and thus constraining their effectiveness in high-fidelity few-shot NVS. Furthermore, they neglect the importance of the initialization of scene representation for 3DGS.

## 3 METHODS

### 3.1 GENERATIVE POINT CLOUD COMPLETION-BASED GAUSSIAN INITIALIZATION

The sparse point cloud used to initialize 3D Gaussians in FSGS (Zhu et al., 2024) from SfM (Schonberger & Frahm, 2016), provides the initial information on the scene's structure and appearance. In particular, the initial Gaussians' means follow the corresponding points' spatial positions. Since sparse views may cause the corresponding point cloud to become considerably less informative, *i.e.*, *incomplete* regarding the scene's structural representation in under-observed regions.

A straightforward solution is to generate points for completion, which often results in a dilemma: generative models fill structural hollows, but also introduce significant outliers due to unconstrained *hallucination*. As shown in Figure 3 (b), the Gaussians initialized using such points cause structural distortion in those regions with details and degrade the few-shot NVS performance.

Hence, as shown in Figure 2, our unified **G**enerative point cloud **C**ompletion-based **G**aussian **I**nitialization (GCGI) strategy produces refined complementary points to enhance the representation of initial point cloud. Specifically, GCGI comprises two sequential modules on complementary point generation and filtering with the *generate-and-filter* paradigm.

#### 3.1.1 COMPLEMENTARY POINT GENERATION

Inspired by previous studies (Yu et al., 2021b), we design an end-to-end complementary point generation (CPG) module to produce a complementary set of points for point cloud completion.

Given the point cloud $\mathbf{P}_0 = \{p_1, p_2, \ldots, p_n\}$ that has been used to initialize a provisional set of 3D Gaussians $\Theta_0 = \{\theta_1, \theta_2, \ldots, \theta_n\}$, the CPG module starts by using the furthest point sampling (FPS) algorithm (Eldar et al., 1997) to downsample $\mathbf{P}_0$ for a set of point proxies $C_0 = \{c_1, c_2, \ldots, c_n\}$, and adopts a light-weight backbone (*i.e.*, DGCNN (Wang et al., 2019)) $\mathcal{F}$ to extract a representation for each point proxy $c_i$ that represents the corresponding local structural details, as follows:

$$f_i = \mathcal{F}(c_i) + PE(c_i), \tag{1}$$

where $PE(c_i)$ denotes the position embedding of proxy $c_i$.

To exploit the structural representations and the long-range dependencies among different local parts of the point cloud, the CPG module leverages the Transformer model (Vaswani et al., 2017) that comprises an encoder $\mathcal{M}_E$ and a decoder $\mathcal{M}_D$ for the set-to-set generation. In particular, the $k$-NN algorithm (Kramer, 2013) is employed in each Transformer block to capture the structural relationships among point proxies for the enhancement of the local geometric information, *i.e.*, each query representation is enhanced by processing it and its $k$ nearest representations altogether using a linear layer followed by the max pooling operation. The encoder $\mathcal{M}_E$ outputs a set of high-level representations $F'$ from $F = \{f_1, f_2, \ldots, f_n\}$, as follows:

$$F' = \mathcal{M}_E(F). \tag{2}$$

Following the idea of dynamic query mechanism (Dai et al., 2021), the decoder $\mathcal{M}_D$ takes as input both $F'$ and dynamic queries $Q = \{q_1, q_2, \ldots, q_m\}$, and generates a new set of point proxies $C_1 = \{c'_1, c'_2, \ldots, c'_m\}$, as follows:

$$C_1 = \mathcal{M}_D(F', Q). \tag{3}$$

Afterward, the CPG employs a point auto encoder-decoder $\mathcal{H}$, *i.e.*, FoldingNet (Yang et al., 2018), to output a set of complementary points $\mathbf{P}_1 = \{P'_1, P'_2, \ldots, P'_m\}$ with structural details, as follows,

$$P'_i = \mathcal{H}(c'_i), \tag{4}$$

where $P'_i$ denotes the neighboring points centered at $c'_i$.

### 3.1.2 COMPLEMENTARY POINT FILTERING

As shown in Figure 3 (b), the combined point cloud $\mathbf{P}_c = \mathbf{P}_0 \bigcup \mathbf{P}_1$ with the naive combination of sparse point cloud $\mathbf{P}_0$ and the complementary points $\mathbf{P}_1$ output by CPG module contains significant outliers.

To address this points generative *hallucination*, we devise an additional complementary point filtering (CPF) module to prune outliers in $\mathbf{P}_1$ while maintaining the scene's structural details.

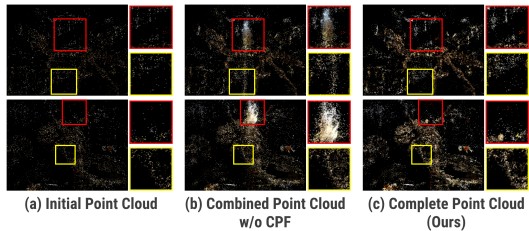

(a) Initial Point Cloud    (b) Combined Point Cloud w/o CPF    (c) Complete Point Cloud (Ours)

Figure 3: Comparison of initial point cloud $\mathbf{P}_0$, combined point cloud $\mathbf{P}_c$, and final complete point cloud $\mathbf{P}_f$.

Previous studies have demonstrated that structures like anchor grids or octrees can contribute to enhancing the local structural details for 3DGS (Lu et al., 2024; Ren et al., 2025). Since few-shot NVS is an ill-posed problem, introducing additional structural information that needs to be optimized would cause training to crash. Therefore, we design a filtering mask in the CPF module to detect outliers for pruning based on K-Dimensional Tree (kd-Tree) (Zhou et al., 2008), an optimize-free space-partitioning data structure.

In the absence of ground truth structural information, the incomplete point cloud $\mathbf{P}_0$ initially obtained through the SfM is used as a high-confidence reference, for which the CPF module constructs a kd-tree $\mathcal{T} = \{t_1, t_2, \ldots, t_d\}$ that comprises $d$ parts using the nearest-neighbor search algorithm. For each complementary point $p'_i \in \mathbf{P}_1$, the CPF module samples $k = 3$ nearest points $\{p_{i,1}, p_{i,2}, \ldots, p_{i,k}\} \in \mathbf{P}_0$, as reference anchors, in its corresponding part $t_i$ of $\mathcal{T}$, as follows,

$$p_{i,k} = k\text{-}min_{p \in (\mathbf{P}_0 \cap t_i)} \|p'_i - p\|, \quad p'_i \in t_i. \tag{5}$$

The reference anchors are adopted to calculate a distance-based outlier indicator $y_i$ for $p'_i$ as follows,

$$y_i = \frac{1}{k} \sum_{i=1}^{k} \|p'_i - p_{i,k}\|. \tag{6}$$

Afterward, the CPF module conducts a binary classification on each complementary point $p'_i \in \mathbf{P}_1$, *i.e.*, $p'_i$ as an outlier if $y_i$ exceeds a predefined threshold $\delta_1 = 1.0$ and the mean distance of $\mathbf{P}_0$. Accordingly, the filtering mask is obtained as follows,

$$M = \mathbf{1}(y \leq \delta_1 \cdot \mu(\mathbf{P}_0)) \quad \mu(\mathbf{P}_0) = \frac{1}{n(n-1)} \sum_{i=1}^{n-1} \sum_{j \neq i}^{n} \|p_i - p_j\|. \tag{7}$$

The module then leverages this mask to filter those points distant to high-confidence reference $\mathbf{P}_0$ from $\mathbf{P}_1$. And, the complete point cloud $\mathbf{P}_f$, which possesses enhanced structural information barely affected by outliers.

$$\mathbf{P}'_1 = \mathbf{P}_1 \odot M, \quad \mathbf{P}_f = \mathbf{P}_0 \cup \mathbf{P}'_1. \tag{8}$$

Given the complete point cloud $\mathbf{P}_f$, a set of 3D Gaussians $\Theta$ for optimization can be initialized as follows,

$$\Theta = \Theta_0 \cup \Theta_1, \tag{9}$$

where $\Theta_1$ represents the complementary 3D Gaussians initialized using $\mathbf{P}'_1$. Specifically, the position of each Gaussian $\theta'_i \in \Theta_1$ follows a point $p'_i \in \mathbf{P}'_1$, whereas the remaining attributes of $\theta'_i$ are cloned from those of Gaussian in $\Theta_0$ corresponding to nearest point $p_j \in \mathbf{P}_0$ of $p'_i$ according to $\mathcal{T}$.

### 3.2 GENERATIVE PSEUDO VIEW COMPLETION-BASED GAUSSIAN OPTIMIZATION

To exploit the sparse training views while preventing overfitting, existing methods (Zhu et al., 2024; Zhang et al., 2024) have attempted to employ pseudo views generated from interpolated camera poses as additional guidance for training. Since such pseudo views are essentially based on the observed regions of the scene, this strategy often still causes *hollows* or incomplete structural details in the reconstruction of those regions unobserved by the input training views after Gaussian optimization. As a countermeasure, our GenCoGS adopts a **G**enerative point cloud **Co**mpletion-based **G**aussian **O**ptimization (GCGO) strategy based on an I2V diffusion model (Yu et al., 2024a) in Figure 2, which is capable of maintaining spatial-temporal consistency, for structurally-aware pseudo view completion against hollows.

Specifically, the input training views are processed by the image encoder of a pre-trained language-image model, *e.g.*, CLIP (Radford et al., 2021), to obtain high-level representations $F_c$ that hold multi-view consistency information. These representations are then integrated with each initial pseudo view $I_p$ to provide the conditional information that guides the diffusion model to reach the corresponding complete pseudo view $\hat{I}_p$ via a multi-step denoising process, as follows:

$$z_{t-1} = p_\theta(z_t, \mathbb{E}[z_0 \mid z_t, F_c, I_p]), \quad \hat{I}_p = \mathcal{G}(z_T), \tag{10}$$

where $p_\theta$ denotes the denoising process, $z_t$ denotes the high-level representations from VAE of LDMs (Metzer et al., 2023) at denoising step $t$, $\mathcal{G}$ refers to the image generator and $T$ denotes the final step. Please refer to **Preliminary in Appendix** for detils.

#### 3.2.1 PERTURBED CAMERA TRAJECTORY

To explore those unobserved regions of the scene by the input training views, we introduce a perturbed camera trajectory that benefits pseudo camera pose sampling. Specifically, uniform poses are first sampled in a circular camera trajectory generated from the camera poses of training views (Ovrén & Forssén, 2018) as candidate pseudo camera pose positions. Afterward, each pseudo pose $\mathbf{c}_i$ is defined by a position $t_i$ and a quaternion on the rotation $q_i$ averaged from two training cameras. In particular, our strategy applies periodic perturbations alongside the x-axis and y-axis of the camera coordinate system using the $sin$ function to it, which may cover horizontally and vertically distributed unobserved regions, as follows:

$$\mathbf{c}_i = [t_i + Asin(2\pi f \cdot t_i) \begin{bmatrix} 1 \\ 1 \\ 0 \end{bmatrix}, q_i], \tag{11}$$

where $A$ represents the x-axis and y-axis perturbation amplitudes, and $f$ denotes the wave frequency. We set $f = 1.0$, and $A = 2.0$ as the trade-off between exploiting unobserved regions and avoiding generative model hallucination.

#### 3.2.2 GENERATIVE CONSISTENCY LOSS

Similar to the generative points, the generative hallucination in the complete pseudo views $\hat{I}_p$ could result in multi-view inconsistency and appearance distortion in the rendered details, as shown in Figure 4. To attenuate this impact, we design a generative consistency loss composed of two key terms on constraining those regions' representations with appearance distortion and improving the scene completion capability while maintaining the multi-view consistency.

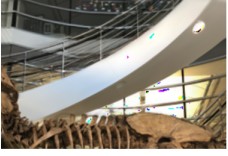
**(a) Synthesized View**

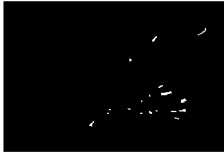
**(b) Confidence Mask**

Figure 4: Hallucination-caused appearance distortion in a synthesized view and corresponding confidence mask $\hat{M}_r$.

Specifically, the first loss term is based on a pixel-level confidence mask $M_r$, which firstly evaluates the appearance gap $\Delta_C$ between the color $C$ of $I_p$ and $\hat{I}_p$ via the L2-norm, formulated for a pixel $(u, v)$ as follows,

$$\Delta_C(u, v) = \|C_{I_p}(u, v) - C_{\hat{I}_p}(u, v)\| \tag{12}$$

Subsequently, we generated an adaptive threshold $T(u, v)$ to robustly identify significant distortion. Specifically, a Gaussian blur kernel is adopted to generate the local mean $\mu_\Delta(u, v)$ and standard deviation $\sigma_\Delta(u, v)$ as the local statistics of the gap $\Delta_C$, and $T(u, v)$ are derived as follows,

$$T(u, v) = \mu_\Delta(u, v) + \delta_2 \cdot \sigma_\Delta(u, v), \tag{13}$$

where $\delta_2 = 20$ denotes as a variance coefficient. Finally, the binary confidence mask $M_r$ is obtained by applying the adaptive threshold to the difference map:

$$M_r(u, v) = \begin{cases} 1 & \text{if } \Delta_C(u, v) > T(u, v), \\ 0 & \text{otherwise.} \end{cases} \tag{14}$$

To further improve the coherence and smoothness of $M_r$ for training stability, a sequence of expansion $\mathcal{K}_1$, erosion $\mathcal{K}_2$, and connected components filtering operations is performed as follows:

$$M_r' = (M_r \oplus \mathcal{K}_1) \ominus \mathcal{K}_2, \quad \hat{M}_r = \bigcup_{R_i \in \mathcal{R} \mid \text{Area}(R_i) \geq \delta_3} R_i, \tag{15}$$

where $\mathcal{R}$ denotes the set of connected components in $M_r'$, and $\delta_3 = 8$ refers to a threshold.

Afterward, the first loss term is formulated to constrain the appearance of those regions identified by $\hat{M}_r$ and suppress the hallucination using the $L1$ loss as follows,

$$\mathcal{L}_{reg}(I_p, \hat{I}_p) = \|I_p - \hat{I}_p\|_1 \odot \hat{M}_r, \tag{16}$$

The second loss term provides a feature-level constraint between $I_p$ and $\hat{I}_p$ based on a VGG network Simonyan & Zisserman (2015), to benefit structural completion and keep multi-view consistency, as follows:

$$\mathcal{L}_{str}(I_p, \hat{I}_p) = \mathcal{L}_{LPIPS}(I_p, \hat{I}_p). \tag{17}$$

Hence, generative consistency loss is formulated with the weight coefficient $\alpha = 10.0$ as follows,

$$\mathcal{L}_{GC} = \mathcal{L}_{img} + \alpha(\mathcal{L}_{reg} + \mathcal{L}_{str}), \tag{18}$$

where, $\mathcal{L}_{img}$ represents reconstruction loss Kerbl et al. (2023) between $I_p$ and $\hat{I}_p$ using $\lambda = 0.2$,

$$\mathcal{L}_{img}(I_p, \hat{I}_p) = \mathcal{L}_1(I_p, \hat{I}_p) + \lambda \mathcal{L}_{DSSIM}(I_p, \hat{I}_p). \tag{19}$$

### 3.2.3 GAUSSIAN OPTIMIZATION

The proposed method adopts a two-phase optimization for 3D Gaussians. At the first phase, *i.e.*, during the first $m$ iterations, Gaussians are optimized solely using an image reconstruction loss $\mathcal{L}_{img}$ between the synthesized views and training views; At the second phase, *i.e.*, during the following iterations, pseudo camera poses are sampled based on our camera trajectory perturbation strategy to generate the corresponding pseudo views, which contribute to the optimization of 3D Gaussians. Overall, the training loss is formulated as follows,

$$\mathcal{L} = \begin{cases} \mathcal{L}_{img}, & \text{if } k < m, \\ \mathcal{L}_{img} + \beta \mathcal{L}_{GC}, & \text{otherwise,} \end{cases} \tag{20}$$

where $k$ represents the iteration index and $\beta$ denotes a weight coefficient. We set $\beta = 0.1$ in practice.

Table 1: Comparison of GenCoGS and other methods regarding few-shot NVS performance on the LLFF (Mildenhall et al., 2019) dataset under 3-view, 6-view and 9-view settings. The best , second-best , and third-best scores are highlighted.

| Method | PSNR↑ | | | SSIM↑ | | | LPIPS↓ | | | AVGE↓ | | |
|---|---|---|---|---|---|---|---|---|---|---|---|---|
| | 3 | 6 | 9 | 3 | 6 | 9 | 3 | 6 | 9 | 3 | 6 | 9 |
| SparseNeRF (Wang et al., 2023) | 19.86 | 23.26 | 24.27 | 0.714 | 0.741 | 0.781 | 0.243 | 0.235 | 0.228 | 0.127 | 0.117 | 0.113 |
| ReconFusion (Wu et al., 2024) | 21.34 | 24.25 | 25.21 | 0.724 | 0.815 | 0.848 | 0.203 | 0.152 | 0.134 | 0.110 | 0.090 | 0.081 |
| MuRF (Xu et al., 2024) | 21.26 | 23.54 | 24.66 | 0.722 | 0.796 | 0.836 | 0.245 | 0.199 | 0.164 | 0.118 | 0.103 | 0.094 |
| FrugalNeRF (Lin et al., 2025) | 19.87 | - | - | 0.610 | - | - | 0.300 | - | - | 0.125 | - | - |
| CAT3D (Gao et al., 2024) | 21.58 | 24.71 | 25.63 | 0.731 | 0.833 | 0.860 | 0.181 | 0.121 | 0.107 | 0.097 | 0.067 | 0.059 |
| 3DGS (Kerbl et al., 2023) | 15.52 | 19.45 | 21.13 | 0.405 | 0.627 | 0.715 | 0.408 | 0.268 | 0.214 | 0.209 | 0.154 | 0.137 |
| FSGS (Zhu et al., 2024) | 20.31 | 24.20 | 25.32 | 0.652 | 0.811 | 0.856 | 0.288 | 0.173 | 0.136 | 0.136 | 0.095 | 0.082 |
| DNGaussian (Li et al., 2024a) | 19.12 | 22.18 | 23.17 | 0.591 | 0.755 | 0.788 | 0.294 | 0.198 | 0.180 | 0.132 | 0.110 | 0.105 |
| BinoGS (Han et al., 2024) | 21.44 | 24.87 | 26.17 | 0.751 | 0.845 | 0.877 | 0.168 | 0.106 | 0.090 | 0.101 | 0.061 | 0.051 |
| IPSM (Wang et al., 2024) | 20.44 | 23.91 | 25.13 | 0.702 | 0.818 | 0.855 | 0.207 | 0.135 | 0.111 | 0.109 | 0.080 | 0.071 |
| ReconX (Liu et al., 2025) | 21.05 | - | - | 0.768 | - | - | 0.178 | - | - | 0.111 | - | - |
| **GenCoGS (Ours)** | 22.13 | 25.61 | 26.64 | 0.762 | 0.857 | 0.880 | 0.164 | 0.108 | 0.090 | 0.084 | 0.051 | 0.044 |

Table 2: Comparison of GenCoGS and other methods regarding performance on the DTU (Jensen et al., 2014) under 3-view setting.

| Method | PSNR↑ | SSIM↑ | LPIPS↓ | AVGE↓ |
|---|---|---|---|---|
| SparseNeRF (Wang et al., 2023) | 19.47 | 0.829 | 0.183 | 0.120 |
| ReconFusion (Wu et al., 2024) | 20.74 | 0.875 | 0.124 | 0.109 |
| MuRF (Xu et al., 2024) | 21.31 | 0.885 | 0.127 | 0.103 |
| CAT3D (Gao et al., 2024) | 22.02 | 0.844 | 0.121 | 0.099 |
| FSGS (Zhu et al., 2024) | 17.34 | 0.818 | 0.169 | 0.123 |
| DNGaussian (Li et al., 2024a) | 18.91 | 0.790 | 0.176 | 0.124 |
| BinoGS (Han et al., 2024) | 20.71 | 0.862 | 0.111 | 0.096 |
| IPSM (Wang et al., 2024) | 19.99 | 0.856 | 0.121 | 0.077 |
| ReconX (Liu et al., 2025) | 19.78 | 0.476 | 0.378 | 0.142 |
| **GenCoGS (Ours)** | 23.11 | 0.910 | 0.082 | 0.049 |

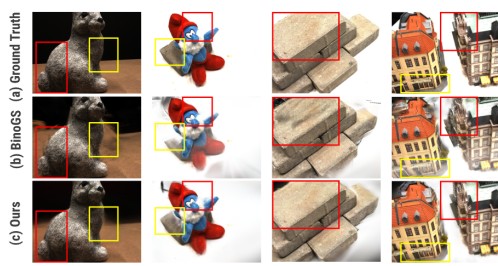

Figure 5: Visualization of example by GenCoGS and BinoGS (Han et al., 2024) on DTU dataset under 3-view setting.

## 4 EXPERIMENTS

Following previous methods (Zhu et al., 2024; Paliwal et al., 2024), we conducted experiments on three benchmark datasets: LLFF (Mildenhall et al., 2019), DTU (Jensen et al., 2014), and Shiny (Wizadwongsa et al., 2021) with 3, 6, and 9 training views as few-shot settings. We implemented GenCoGS using the PyTorch framework, with the initial point cloud computed from SfM in FSGS (Zhu et al., 2024). During optimization, we densify the Gaussians every 100 iterations and start densification after 1000 iterations. The total optimization steps are set to 5000, and we set the *GCGO* after $m = 4,000$ iterations. For hyper-parameters, we set $k = 3$ and $\delta_1 = 1.0$ in *GCGI*, we set the wave frequency $f = 1.0$, perturbation amplitude $A = 2.0$, $\delta_2 = 20$, and $\delta_3 = 8$ in *GCGO*, and the loss weight coefficients are set as $\alpha = 10.0$, and $\beta = 0.1$ for Gaussian optimization. All results are obtained using a NVIDIA A6000 GPU. Furthermore, please refer to the **Appendix for details on Datasets and Evaluation Metrics**.

### 4.1 QUANTITATIVE COMPARISON

As shown in Table 1, 2 and 3, our GenCoGS consistently outperformed other representative few-shot NVS methods, nearly in all metrics. On the LLFF dataset, GenCoGS achieved improvements of 0.55 dB / 0.74 dB / 0.47 dB in PSNR, 0.011 / 0.012 / 0.003 in SSIM, and 0.013 / 0.029 / 0.027 in AVGE under 3-view / 6-view / 9-view settings, respectively, compared to the methods with second-best performances. On the DTU dataset, the improvements by GenCoGS under 3-view set-

Table 3: Comparison of GenCoGS and other methods regarding performance on the Shiny (Jensen et al., 2014) under 3-view setting.

| Method | PSNR↑ | SSIM↑ | LPIPS↓ | AVGE ↓ |
|---|---|---|---|---|
| RegNeRF | 18.10 | 0.574 | 0.378 | 0.136 |
| FreeNeRF | 18.65 | 0.586 | 0.360 | 0.127 |
| SparseNeRF | 18.81 | 0.591 | 0.354 | 0.124 |
| 3D-GS | 17.83 | 0.547 | 0.385 | 0.142 |
| FSGS | 19.63 | 0.612 | 0.327 | 0.111 |
| **GenCoGS (Ours)** | 21.10 | 0.692 | 0.202 | 0.099 |

ting were 2.40 dB in PSNR, 0.025 in SSIM, 0.029 in LPIPS, and 0.045 in AVGE compared to the second-best 3DGS-based method. Please refer to **Appendix** for detailed results on the DTU dataset. Notably, the substantial boosts over other diffusion-based methods (Wang et al., 2024; Wu et al.,

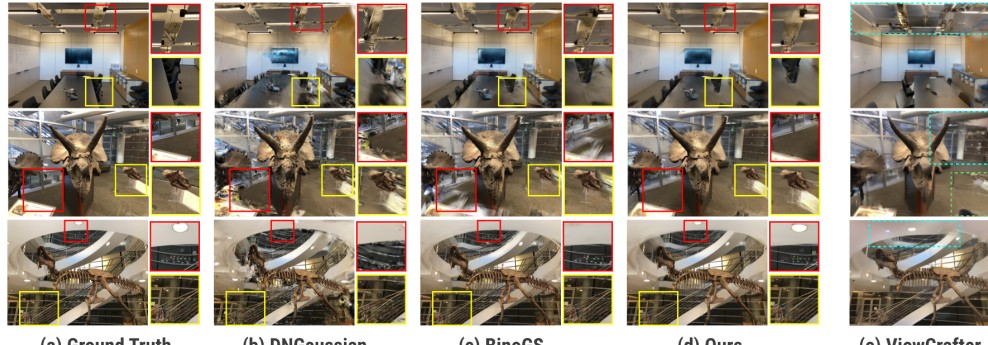

|  |  |  |  |  |
|---|---|---|---|---|
| (a) Ground Truth | (b) DNGaussian | (c) BinoGS | (d) Ours | (e) ViewCrafter |

Figure 6: Visualization of example synthesized views by GenCoGS, DNGaussian (Li et al., 2024a), BinoGS (Han et al., 2024) and ViewCrafter (Yu et al., 2024a) on LLFF under the 3-view setting.

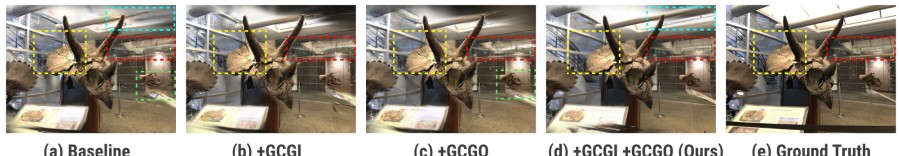

|  |  |  |  |  |
|---|---|---|---|---|
| (a) Baseline | (b) +GCGI | (c) +GCGO | (d) +GCGI +GCGO (Ours) | (e) Ground Truth |

Figure 7: Visualization of example images in the ablation studies on LLFF under 3-view setting.

2024; Gao et al., 2024) achieved by GenCoGS stem from the hallucination attenuation capability of our strategies.

On the more challenging Shiny dataset, GenCoGS also outperformed existing methods, achieving improvements of 1.47 dB in PSNR, 0.080 in SSIM, 0.125 in LPIPS, and 0.012 in AVGE under the 3-view setting, which further validates the superiority of GenCoGS in high-fidelity few-shot NVS.

### 4.2 QUALITATIVE COMPARISON

We visualized example views synthesized by GenCoGS, alongside DNGaussian (Li et al., 2024a), BinoGS (Han et al., 2024) and the diffusion-based ViewCrafter (Yu et al., 2024a), on both DTU and LLFF datasets under 3-views setting, as shown in Figure 5 and 6.

DNGaussian and BinoGS attempted to exploit priors on the structure and appearance of the input training views, but resulted in considerable ambiguity, *e.g.*, first and second rows in Figure 6, due to the lack of scene completion capability. Furthermore, the results of ViewCrafter (Yu et al., 2024a) suggest that its generative completion pipeline toward synthesized views suffers from significant generative model hallucination and unsatisfactory scene reconstruction capability via synthesized view completion, as shown in Figure 6 (e) highlighted regions.

Integrating both generative completion-based strategies, our GenCoGS provided a high-quality scene using the complete initial Gaussians followed by the optimization additionally guided by pseudo views less influenced by generative model hallucination. In particular, our GCGO strategy effectively filled the hollows within the synthesized views, *e.g.*, the highlighted regions in Figure 6 second and third rows. These examples further demonstrate the improvements of GenCoGS across different benchmark datasets, demonstrating its consistency and effectiveness in delivering high-fidelity few-shot NVS results.

### 4.3 ABLATION STUDIES

To investigate the contributions of individual strategies, we conducted ablation studies on the LLFF dataset under the 3-view setting. The results in Table 4 indicate that each strategy positively impacts few-shot NVS performance, with the combination of both achieving the best performance.

**Impact of the GCGI Strategy** Compared to the baseline, adopting the GCGI strategy reached the improvements of 0.66 dB, 0.024, 0.016 and 0.009 in PSNR, SSIM, LPIPS and AVGE, respectively. These results suggest that the complete initial Gaussians with the complete point cloud from the

Table 4: Ablation of our GCGI and GCGO strategies on LLFF under 3-view.

|  | PSNR | SSIM | LPIPS | AVGE |
|---|---|---|---|---|
| Baseline | 20.79 | 0.733 | 0.184 | 0.096 |
| + GCGI | 21.45 | 0.757 | 0.168 | 0.087 |
| + GCGO | 21.65 | 0.752 | 0.184 | 0.088 |
| + GCGI + GCGO (Ours) | 22.13 | 0.762 | 0.164 | 0.084 |

Table 5: Ablation of pseudo camera sampling and $\mathcal{L}_{GC}$ in GCGO on LLFF under 3-view.

| Sampling | $\mathcal{L}_{GC}$ | PSNR | SSIM | LPIPS |
|---|---|---|---|---|
| Random | ✓ | 21.83 | 0.755 | 0.188 |
| Camera Trajectory | ✗ | 21.59 | 0.749 | 0.181 |
| Camera Trajectory | ✓ | 22.13 | 0.762 | 0.164 |

Figure 8: Visualization of pseudo views by I2V diffusion model using different $A$.

Table 6: Ablation of our CPG and CPF modules in the GCGI strategy on LLFF under 3-view. $1/4$ means random sampling a quarter of $\mathbf{P}_0$.

| Sampling | w/ CPG | w/ CPF | PSNR | SSIM | LPIPS |
|---|---|---|---|---|---|
| Full |  |  | 21.65 | 0.752 | 0.184 |
| Full | ✓ |  | 22.04 | 0.760 | 0.178 |
| Full | ✓ | ✓ | 22.13 | 0.762 | 0.164 |
| 1/4 |  |  | 21.24 | 0.730 | 0.199 |
| 1/4 | ✓ |  | 21.61 | 0.733 | 0.195 |
| 1/4 | ✓ | ✓ | 21.78 | 0.741 | 0.191 |

GCGI strategy is capable of avoiding floating artifacts in those scene regions with details, as also illustrated in Figure 7. As shown in Figure 3, our CPG and CPF modules work jointly to refine the sparse initial point cloud into a more complete one while effectively removing outliers to avoid hallucination.

As shown in Table 6, both modules consistently contributed to improvements even when the quality of the initial point cloud $\mathbf{P}_0$ was degraded by randomly sampling only a quarter of the points. This demonstrates the strong generalization capability and robustness of our GCGI strategy.

**Impact of the GCGO Strategy** Compared to the baseline, leveraging the GCGO strategy achieved the improvements of 0.86 dB, 0.019, and 0.008 in PSNR, SSIM, and AVGE, respectively. As illustrated in Figure 7, Gaussians optimized using the GCGO strategy mitigated hollows and floating artifacts, which benefited the synthesis of high-fidelity views.

In particular, as shown in Table 5, the pseudo camera poses sampled from a perturbed camera trajectory facilitated better scene completion in unobserved regions compared to randomly sampled poses. It is noteworthy that our $\mathcal{L}_{GC}$ further improved performance by focusing on reducing generative model hallucination.

Furthermore, we identified a critical see-saw effect between generative model hallucination and unobserved region exploration based on the perturbed camera trajectory. As shown in Figure 8, the I2V model generated significant hallucination when trying to cover more unobserved regions, leading to low-fidelity outcomes. Hence, we set $A = 2.0$ as a balanced trade-off in our experiments. Furthermore, please kindly refer to **Appendix** for additional experiments results.

## 5 CONCLUSIONS

In this paper, we addressed a critical limitation of existing 3DGS-based few-shot NVS methods, *i.e.*, unsatisfactory scene completion capability caused by the overdependence on the observed regions of sparse training views. Our unified method, GenCoGS, enhances scene completion by incorporating two generative completion-based strategies focusing on Gaussian initialization and optimization. For Gaussian initialization, GenCoGS generates and filters complementary points to establish a complete point cloud with refined structural and appearance information; For Gaussian optimization, GenCoGS leverages an image-to-video (I2V) diffusion model to generate complete pseudo views, providing effective guidance over unobserved scene regions while attenuating generative model hallucination. By enabling accurate and coherent scene completion, GenCoGS outperformed representative 3DGS-based few-shot NVS methods and achieved significant improvements, demonstrating the superiority of GenCoGS.

## REPRODUCIBILITY STATEMENT

To ensure the reproducibility of our work, we provided comprehensive details on our methodology and experiments. The motivation and architectural design of our proposed strategies are elaborated in Section 3. A complete description of the experiments implementation, including all hyperparameter configurations, was provided in Section 4. To justify our hyperparameter choices, we also present extensive ablation studies. The source code will be made publicly available under an open-source license upon the acceptance of this paper. We also performed the video qualitative visualizations in Supplementary Materials, please kindly refer to them for comparison with other methods.

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

APPENDIX

# A  PRELIMINARY

## A.1  3D GAUSSIANS SPLATTING

3DGS Kerbl et al. (2023) uses a set of 3D Gaussians $\theta$ for scene representation and achieves a 2D image with pixel-wise color C from a view's pose. The $i$-th Gaussian is formulated as $\theta_i = \{\mu_i, S_i, R_i, \alpha_i, f_i\}$, where $\mu_i \in \mathbb{R}^3$ is the mean (*i.e.*, position), $S_i \in \mathbb{R}^{3\times3}$ is the scaling matrix, $R_i \in \mathbb{R}^{3\times3}$ is the rotation matrix, $\alpha_i \in \mathbb{R}$ is the opacity, and $f_i \in \mathbb{R}^K$ is the $K$-dimensional color representation. The basis function of $\theta_i$ on the position $x$ is defined with the covariance matrix $\Sigma_i \in \mathbb{R}^{3\times3}$ as follows,

$$\mathrm{G}_i(x) = e^{-\frac{1}{2}(x-\mu_i)'\Sigma_i^{-1}(x-\mu_i)}, \quad \Sigma_i = R_i S_i S_i' R_i', \tag{21}$$

where $R_i$ is an orthogonal matrix and $S_i$ is a diagonal matrix.

During the rasterization process, $\theta$ are splatted onto an image plane via the projection along the depth dimension. For an image pixel $x_p$, its color $\mathrm{C}(x_p)$ is the result of the $\alpha$-blending of $N$ ordered Gaussians that intersect with the ray for $x_p$, as follows:

$$\mathrm{C}(x_p) = \sum_{i\in N} c_i \alpha_i T_i, \quad T_i = \prod_{j=1}^{i-1}(1-\alpha_j), \tag{22}$$

where $c_i$ is the color calculated from the spherical harmonic (SH) coefficients of $f_i$, and $T_i$ is the corresponding accumulated transmittance at $x_p$. Different from the ray sampling strategy in NeRF Mildenhall et al. (2020), these Gaussians are hit by a parallelized rasterizer according to $x_p$ and $P$.

## A.2  IMAGE-TO-VIDEO DIFFUSION MODEL

Diffusion model consists of two primary components Song et al. (2020): a forward process $q$ and a reverse process $p_\theta$. The forward process gradually introduces noise to clean data $x_0$, creating a noisy state $x_t = \alpha_t x_0 + \sigma_t \epsilon$ (where $\epsilon \sim \mathcal{N}(0,\mathbf{I})$ and $\alpha_t^2 + \sigma_t^2 = 1$) across different time steps. The reverse process $p_\theta$ focuses on denoising from the noisy data to clean data distribution utilizing a noise predictor $\epsilon_\theta$, which is optimized by the objective:

$$\min_\theta \mathbb{E}_{t\sim\mathcal{U}(0,1),\epsilon\sim\mathcal{N}(0,\mathbf{I})} \left[ \|\epsilon_\theta(x_t,t) - \epsilon\|_2^2 \right]. \tag{23}$$

In image-to-video (I2V) diffusion, Latent Diffusion Models (LDMs) Metzer et al. (2023) are employed in a compact latent space for efficiency. The data $x \in \mathbb{R}^{L\times3\times H\times W}$ is encoded into the latent space by a pretrained VAE $z = \mathcal{E}(x)$, $z \in \mathbb{R}^{L\times C\times h\times w}$ frame-by-frame. Then, both the forward process $q$ and the reverse process $p_\theta$ are performed in the latent space. The final generated videos are obtained through the VAE decoder $\hat{x} = \mathcal{D}(z)$. In this work, we build our model based on an open-sourced I2V diffusion model ViewCrafter Yu et al. (2024a). This aligns naturally with our goal of NVS from sparse views.

# B  EXPERIMENTS

## B.1  EXPERIMENTS SETTING

### B.1.1  DATASETS

We conducted experiments on three benchmark datasets: Local Light Field Fusion (LLFF) Mildenhall et al. (2019), characterized by forward-facing scenes; DTU Jensen et al. (2014), featuring object-centric scenes; and Shiny Wizadwongsa et al. (2021), including challenging scenes with view-dependent effects. Following previous studies Niemeyer et al. (2022); Zhu et al. (2024); Paliwal et al. (2024), we adopted the same split strategy over all the datasets. For the LLFF and DTU datasets, we applied the downsampling rate of 8 and that of 4, respectively, and used few-shot settings with 3, 6 and 9 training views; for each scene, the evaluation set is fixed regardless of the

Table 7: The additional comparison of GenCoGS and other methods regarding few-shot NVS performance on the DTU (Jensen et al., 2014) dataset under 3-view, 6-view and 9-view settings. The best , second-best , and third-best scores are highlighted.

| Method | PSNR↑ | | | SSIM↑ | | | LPIPS↓ | | | AVGE↓ | | |
|---|---|---|---|---|---|---|---|---|---|---|---|---|
| | 3 | 6 | 9 | 3 | 6 | 9 | 3 | 6 | 9 | 3 | 6 | 9 |
| FreeNeRF (Yang et al., 2023) | 19.52 | 23.25 | 25.38 | 0.787 | 0.844 | 0.888 | 0.173 | 0.131 | 0.102 | 0.119 | 0.096 | 0.082 |
| SparseNeRF (Wang et al., 2023) | 19.47 | - | - | 0.829 | - | - | 0.183 | - | - | 0.120 | - | - |
| ReconFusion (Wu et al., 2024) | 20.74 | 23.62 | 24.62 | 0.875 | 0.904 | 0.921 | 0.124 | 0.105 | 0.094 | 0.109 | 0.086 | 0.080 |
| MuRF (Xu et al., 2024) | 21.31 | 23.74 | 25.28 | 0.885 | 0.921 | 0.936 | 0.127 | 0.095 | 0.084 | 0.103 | 0.082 | 0.075 |
| CAT3D (Gao et al., 2024) | 22.02 | 24.28 | 25.92 | 0.844 | 0.899 | 0.928 | 0.121 | 0.095 | 0.073 | 0.099 | 0.075 | 0.071 |
| 3DGS (Kerbl et al., 2023) | 10.99 | 20.33 | 22.90 | 0.585 | 0.776 | 0.816 | 0.313 | 0.223 | 0.173 | 0.224 | 0.151 | 0.129 |
| FSGS (Zhu et al., 2024) | 17.34 | 21.55 | 24.33 | 0.818 | 0.880 | 0.911 | 0.169 | 0.127 | 0.106 | 0.123 | 0.103 | 0.092 |
| DNGaussian (Li et al., 2024a) | 18.91 | 22.10 | 23.94 | 0.790 | 0.851 | 0.887 | 0.176 | 0.148 | 0.131 | 0.124 | 0.106 | 0.097 |
| BinoGS (Han et al., 2024) | 20.71 | 24.31 | 26.70 | 0.862 | 0.917 | 0.947 | 0.111 | 0.073 | 0.052 | 0.096 | 0.073 | 0.052 |
| IPSM (Wang et al., 2024) | 19.99 | - | - | 0.856 | - | - | 0.121 | - | - | 0.077 | - | - |
| ReconX (Liu et al., 2025) | 19.78 | - | - | 0.476 | - | - | 0.378 | - | - | 0.142 | - | - |
| **GenCoGS (Ours)** | 23.11 | 26.45 | 28.53 | 0.910 | 0.939 | 0.960 | 0.082 | 0.059 | 0.043 | 0.049 | 0.032 | 0.023 |

Table 8: The chamfer distances between SfM (Zhu et al., 2024), complete point cloud $\mathbf{P}_f$ after GCGI and final optimized 3DGS point cloud on LLFF (Mildenhall et al., 2019) dataset.

| Scene | Leaves | Orchids | Fortress | Trex | Room | Fern | Horns | Mean(%) |
|---|---|---|---|---|---|---|---|---|
| SfM | 738.21 | 10.15 | 8.42 | 15.39 | 14.45 | 4.38 | 13.41 | 100 |
| Ours | 675.69 | 9.54 | 7.59 | 14.32 | 9.33 | 4.26 | 12.12 | 88 |

number of training views Zhu et al. (2024). For Shiny dataset, we leveraged the 3-view setting, and set the resolutions to $504 \times 378$. In addition, we used the object masks to erase the background noise and focused on scene objects in the DTU dataset, and assumed that camera poses were known, similar to previous studies Niemeyer et al. (2022); Wang et al. (2023); Zhu et al. (2024).

### B.1.2 EVALUATION METRICS

To evaluate the image quality of synthesized views, we leveraged peak signal-to-noise ratio (PSNR) Korhonen & You (2012), structural similarity index measure (SSIM) Wang et al. (2004) and learned perceptual image patch similarity (LPIPS) Zhang et al. (2018) as key metrics. Besides, we calculated the average error (AVGE) derived from the geometric mean of $10^{-\frac{\text{PSNR}}{10}}$, $\sqrt{1 - \text{SSIM}}$, and LPIPS, for a straightforward comparison Niemeyer et al. (2022).

### B.2 ADDITIONAL EXPERIMENTS

### B.2.1 ADDITIONAL QUANTITATIVE COMPARISON

We performed the additional quantitative comparison about the DTU dataset in Table 7, the improvements by GenCoGS under 3-view / 6-view / 9-view settings were 2.40 dB / 2.14 dB / 1.83 dB in PSNR, 0.025 / 0.018 / 0.017 in SSIM, 0.029 / 0.024 / 0.011 in LPIPS, and 0.045 / 0.041 / 0.029 in AVGE, compared to the 3DGS-based methods with second-best performances. Notably, the substantial boosts over other diffusion-based methods (Wang et al., 2024; Wu et al., 2024; Gao et al., 2024) achieved by GenCoGS stem from the hallucination attenuation capability of our strategies.

### B.2.2 QUANTITATIVE OF THE POINT CLOUD AFTER GCGI STRATEGY

To demonstrate the effectiveness of oure GCGI strategy, we measured the chamfer distances of SfM (Zhu et al., 2024) with final optimized 3DGS point cloud, and the complete point cloud $\mathbf{P}_f$ after GCGI strategy with final optimized 3DGS point cloud, respectively. Notably, due to scale discrepancies in numerical values across different scenes, we employed the maximum normalization and used the percentages as the metric in the mean. As shown in Table 8, the mean chamfer distance of our complete point cloud $\mathbf{P}_f$ on LLFF improved to 12%, which demonstrated the effectiveness of our GCGI strategy.

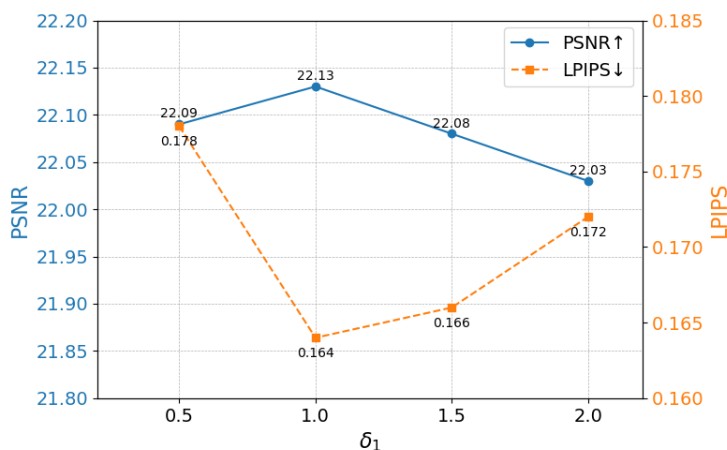

Figure 9: Ablation of the outlier threshold $\delta_1$ in GCGI strategy

Table 9: Ablation of the perturbation amplitude $A$ and the wave frequency $f$ in GCGO.

| Amplitude $A$ | Frequency $f$ | PSNR ↑ | SSIM ↑ | LPIPS ↓ |
|---|---|---|---|---|
| 1.0 | 1.0 | 21.95 | 0.764 | 0.169 |
| 3.0 | 1.0 | 21.83 | 0.757 | 0.171 |
| **(Ours)** 2.0 | 1.0 | **22.13** | **0.762** | **0.164** |
| 2.0 | 0.5 | 22.02 | 0.764 | 0.168 |
| 2.0 | 2.0 | 22.06 | 0.767 | 0.168 |

## B.3 ADDITIONAL ABLATION STUDIES

To investigate the selection of the hyperparameters, we conducted a series of additional ablation studies on the LLFF dataset under the 3-view setting.

### B.3.1 ABLATION OF OUTLIER THRESHOLD $\delta_1$ IN GCGI STRATEGY

We conducted an ablation study of the hyperparameter $\delta_1$ in GCGI strategy on LLFF under the 3-view setting. As shown in Figure 9, we selected $\delta_1 = 1.0$ in the implementation based on the results.

### B.3.2 ABLATION OF $A$ AND $f$ IN GCGO STRATEGY

We conducted an ablation study of the perturbation amplitude $A$ and wave frequency $f$ in the perturbed camera trajectory of GCGO strategy on LLFF under the 3-view setting. As shown in Table 9, there is a critical see-saw effect between generative model hallucination and unobserved region exploration based on $A$ and $f$ as we analyzed in Figure 8. And we selected $A = 2.0$ and $f = 1.0$ as the trade-off.

### B.3.3 ABLATION OF $\delta_2$ IN GCGO STRATEGY

We conducted an ablation study of the coefficient $\delta_2$ in the GCGO strategy on LLFF under the 3-view setting. As shown in Figure 10, the performance improved as $\delta_2$ in GCGO increased, peaking at 20. After this point, performance declined. Consequently, we selected $\delta_2 = 20$ in the implementation based on these observations.

### B.3.4 ABLATION OF $\delta_3$ IN GCGO STRATEGY

We conducted an ablation study of the threshold $\delta_3$ in the GCGO strategy on LLFF under the 3-view setting. As shown in Figure 11, the performance improved as $\delta_3$ in GCGO increased, peaking at 8.

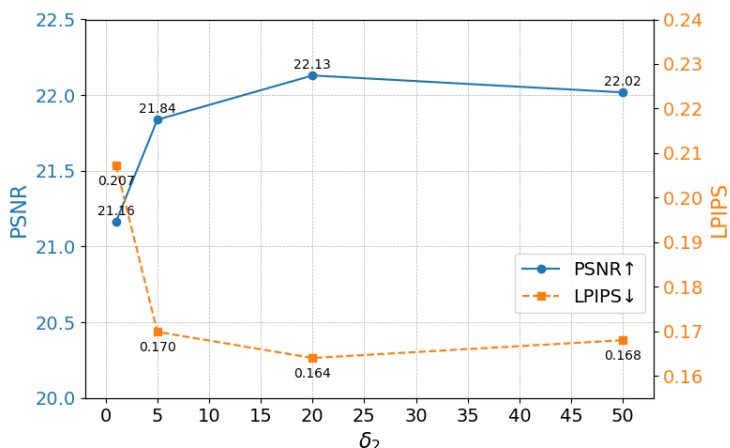

Figure 10: Ablation of the coefficient $\delta_2$ in GCGO strategy

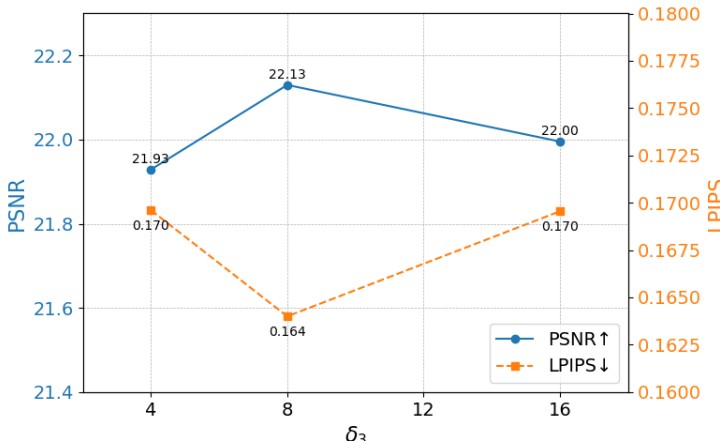

Figure 11: Ablation of the threshold $\delta_3$ in GCGO strategy

After this point, performance declined. Consequently, we selected $\delta_3 = 8$ in the implementation based on these observations.

### B.3.5 ABLATION OF $\beta$ OF $\mathcal{L}_{GC}$ IN GCGO STRATEGY

We conducted an ablation study of the weight coefficient $\beta$ in the GCGO strategy on LLFF under the 3-view setting. As shown in Figure 12, we the selected $\beta = 0.10$ in the implementation for the best performance.

## C LIMITATIONS

Despite the significant improvements in synthesizing high-fidelity views with enhanced scene completion by incorporating two generative completion-based strategies on Gaussian initialization and optimization for few-shot NVS, the proposed method is expected to resolve the following limitations in the future.

- **Computational Efficiency.** We performed the efficiency comparison of our GenCoGS and other representative methods in Table 10, As we discussed in Sec 3 Methods. our GenCoGS was not primarily designed for efficiency, it achieved competitive results across all metrics. Adopting two generative completion-based strategies on Gaussian ini-

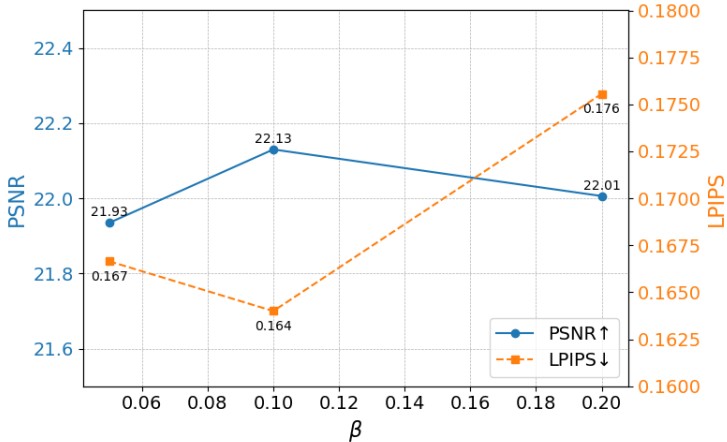

Figure 12: Ablation of the weight coefficient $\beta$ of the $\mathcal{L}_{GC}$ in GCGO strategy

| Method | Inference | | | Training | |
|---|---|---|---|---|---|
| | PSNR ↑ | SSIM ↑ | LPIPS ↓ | Time↓ | Memory↓ |
| FreeNeRF Yang et al. (2023) | 19.63 | 0.612 | 0.308 | 2.3 h | 192.0 GB |
| SparseNeRF Wang et al. (2023) | 19.86 | 0.624 | 0.328 | 1.5 h | 32.0 GB |
| 3DGS Kerbl et al. (2023) | 15.52 | 0.405 | 0.408 | **13.0 min** | **1.6 GB** |
| DNGaussian Li et al. (2024a) | 19.12 | 0.591 | 0.294 | 23.5 min | 2.0 GB |
| FSGS Zhu et al. (2024) | 20.31 | 0.652 | 0.288 | 28.0 min | 2.4 GB |
| BinoGS Han et al. (2024) | 21.44 | 0.751 | 0.168 | 30.0 min | 3.0 GB |
| GenCoGS (Ours) | **22.13** | **0.762** | **0.164** | 40.0 min | 4.0 GB |

Table 10: Comparison of our GenCoGS and other representative methods regarding the efficiency of few-shot NVS on the LLFF Mildenhall et al. (2019) dataset under the 3-view setting.

tialization and optimization, our GenCoGS inevitably increased the training time and memory usage. Although these costs are acceptable compared to performance improvements, we expect to optimize these strategies in the future to enhance computational efficiency. Specifically, the main overhead is due to the denoising process, and future studies could focus on accelerated denoising Esser et al. (2024) as a research direction.

## D THE USE OF LARGE LANGUAGE MODELS (LLMS)

In our work, LLMs **did not** play a significant role in research ideation and/or writing to the extent that they could be regarded as a contributor.

