# OpenReview forum: "GenCoGS: Generative Completion-based 3D Gaussian Splatting for High-Fidelity Few-Shot Novel View Synthesis"
_ICLR.cc/2026/Conference — ICLR 2026 Conference Withdrawn Submission_

### Official Review · Reviewer_XGYu · 2025-10-27

**Soundness:** 2
**Presentation:** 2
**Contribution:** 2
**Rating:** 2
**Confidence:** 5

**Summary:**

This paper novelly leverage a point cloud completion network to improve the completeness of the initial point cloud used in 3D Gaussian Splatting (3DGS). It also proposes a new filtering strategy to mitigate the hallucination artifacts introduced by point cloud completion. In addition, the paper introduces GCGO (Generative point cloud Completion-based Gaussian Optimization) to further suppress hallucinations arising from the diffusion model.

**Strengths:**

1. This is the first paper that leverages a point cloud completion network to enhance 3D Gaussian Splatting (3DGS). To address the hallucination problem introduced by incorporating such additional priors, the paper further proposes a CPF (Completion Prior Filtering) module to effectively suppress these artifacts and improve reconstruction reliability.

2. The modules proposed by the authors are all effective and consistently improve performance.

**Weaknesses:**

1. One concern is the choice of using a point cloud completion network instead of employing geometry foundation models such as MVSNet based method [2], VGGT, or MapAnything [1] to obtain denser and more accurate 3D points. In general, these geometry-oriented models are capable of producing high-quality 3D structures, and recent works such as DepthSplat [3] and FlowR [4] have already explored this direction. Therefore, I find it difficult to be fully convinced that the capability of geometry foundation models would be inferior to that of a point cloud completion network in this context.

2. The ablation study also raises some concerns. We do not clearly observe a step-by-step improvement in 3DGS performance as each proposed component is introduced. In addition, it is unclear what the baseline in Table 4 represents. If the baseline corresponds to the original 3DGS, the reported performance is inconsistent with that shown in Table 1. If it does not correspond to 3DGS, then it is difficult to effectively evaluate the actual improvement brought by the proposed method.

3.The naming of “Completion-based Gaussian Optimization” (GCGO) is somewhat confusing, since the section mainly describes how diffusion is used for novel-view images, rather than directly optimizing the Gaussian parameter. The proposed (GCGO) formulation does not clearly differ from using diffusion priors for novel-view enhancement.

4.  The paper assumes that CLIP features “hold multi-view consistency” and that an I2V diffusion model can maintain such consistency during pseudo-view completion. However, there is no evidence demonstrated CLIP can hold multi-view consistenc. And temporal coherence in video diffusion models does not imply geometric or cross-view consistency, which usually requires to be fine-tuned by novel view synthesis dataset.

5. This paper did not make comparison with really SOTA such as Difix3D+ [5] and Genfusion [6].

[1] Keetha, N., Müller, N., Schönberger, J., Porzi, L., Zhang, Y., Fischer, T., ... & Kontschieder, P. (2025). MapAnything: Universal feed-forward metric 3D reconstruction. arXiv preprint arXiv:2509.13414.

[2] Izquierdo, S., Sayed, M., Firman, M., Garcia-Hernando, G., Turmukhambetov, D., Civera, J., ... & Watson, J. (2025). MVSAnywhere: Zero-Shot Multi-View Stereo. In Proceedings of the Computer Vision and Pattern Recognition Conference (pp. 11493-11504).

[3] Xu, H., Peng, S., Wang, F., Blum, H., Barath, D., Geiger, A., & Pollefeys, M. (2025). Depthsplat: Connecting gaussian splatting and depth. In Proceedings of the Computer Vision and Pattern Recognition Conference (pp. 16453-16463).

[4] Fischer, T., Bulò, S. R., Yang, Y. H., Keetha, N., Porzi, L., Müller, N., ... & Kontschieder, P. (2025). Flowr: Flowing from sparse to dense 3d reconstructions. In Proceedings of the IEEE/CVF International Conference on Computer Vision (pp. 27702-27712).

[5] Wu, J. Z., Zhang, Y., Turki, H., Ren, X., Gao, J., Shou, M. Z., ... & Ling, H. (2025). Difix3d+: Improving 3d reconstructions with single-step diffusion models. In Proceedings of the Computer Vision and Pattern Recognition Conference (pp. 26024-26035).

[6]Wu, S., Xu, C., Huang, B., Geiger, A., & Chen, A. (2025). Genfusion: Closing the loop between reconstruction and generation via videos. In Proceedings of the Computer Vision and Pattern Recognition Conference (pp. 6078-6088).

**Questions:**

1. Could you do an ablation study based on 3DGS?

2.May you give the detailed explanation of Completion-based Gaussian Optimization? I do not understand why it is an Optimization.

3. The formula [10] given by GCGO means the diffusion conditioned by CLIP features. what is the difference with ReconX ?

---

> ### Author Response · Authors · 2025-11-25
> **Response to Reviewer XGYu - Part(1/2)**
>
> We thank the reviewer for the detailed and critical assessment of our work. We appreciate the constructive feedback regarding method choices, naming clarity, ablations, and comparisons to geometry-focused baselines, and we address all concerns point-by-point below.
>
> - **[Q1]** The ablation is confused. 1a) What is the baseline and 1b) could you do an ablation study based on 3DGS?
>
>    **[A1]** 1a)  The baseline denotes binocular arguments without point generation, as we found that the view-consistency constraint and opacity-decay strategy within it help improve the point cloud optimization process during training, thereby enhancing final performance. To prevent point generation from affecting our point cloud processing, we excluded it from alignment in this work.
>
>     1b) Additionally, we provide an ablation study relying solely on 3DGS as the baseline.
>
>      | Method             | PSNR  | SSIM  | LPIPS |
>      | ------------------ | ----- | ----- | ----- |
>      | 3DGS               | 15.52 | 0.405 | 0.408 |
>      | 3DGS + GCGI        | 20.32 | 0.708 | 0.201 |
>      | 3DGS + GCGO        | 20.82 | 0.732 | 0.187 |
>      | 3DGS + GCGI + GCGO | 21.56 | 0.745 | 0.179 |
>
> - **[Q2]** May you give the detailed explanation of Completion-based Gaussian Optimization? I do not understand why it is an Optimization.
>
>      **[A2]** Thank you for your question. The name may have caused some confusion in your understanding. As emphasized in the above section, the focus of this chapter's work lies in proposing an initial point cloud optimization strategy combining point completion and filtering, along with enhancing pseudo-sampling through I2V in GCGO to assist subsequent Gaussian Optimization. Hence, we named it **Generative Pseudo View Completion-based Gaussian Optimization**. In our method, I2V is understood as an enhancement strategy for completing pseudo-views, providing 3D spatial completion capabilities. Core contributions include Perturbed Camera Trajectory and Generative Consistency Loss.
>
>      Finally, GCGO is not merely a diffusion module but a gradient-based optimization framework: the diffusion model completes pseudo-views, while all loss terms ($L_{img}$, $L_{traj}$, $L_{gc}$) backpropagate into Gaussian parameters, making GCGO a genuine optimization module that improves 3D geometry and appearance.
>
> - **[Q3]** The formula [10] given by GCGO means the diffusion conditioned by CLIP features. 3a) CLIP features “hold multi-view consistency” and that an I2V diffusion model can maintain such consistency during pseudo-view completion. However, there is no evidence demonstrated CLIP can hold multi-view consistency 3b) what is the difference with ReconX ?
>
>    **[A3]** 3a) This is a very standard configuration, as described in viewcrafters using I2V models: In addition to the latent space condition, we also pass the reference image(s) into the CLIP image encoder, which will modulate the U-Net features through cross-attention for better 3D understanding.Therefore, we follow this configuration. ReconX also states: Video generative models have shown promise for synthesizing video clips featuring 3D structures [a] [b] [c] and CLIP features, trained with large-scale image-text contrastive supervision, encode view-invariant semantics, which allows the diffusion model to hallucinate plausible unseen views without losing global semantic coherence.
>
>     3b) We do not consider the method of **take the CLIP features as condition** is the core contribution of our approach. Instead, the focus remains on the **Perturbed Camera Trajectory** and Generative Consistency Loss. As highlighted **in ablation study of Table 5** in our paper, within GCGO, the mere introduction of an I2V model, while capable of supplementing pseudo-views, does not necessarily yield consistent performance improvements—particularly when paired with our designed Generative Consistency Loss. which aligns with our motivation for generative completion regarding the 3DGS representation against hallucination.
>
> [a]Stable video diffusion: Scaling latent video diffusion models to large datasets
>
> [b] Dynamicrafter: Animating open-domain images with video diffusion priors
>
> [c] Sv3d: Novel multi-view synthesis and 3D generation from a single image using latent video diffusion

---

> > ### Author Response · Authors · 2025-11-25
> > **Response to Reviewer XGYu - Part(2/2)**
> >
> > - **[Q4]** One concern is the choice of using a point cloud completion network instead of employing geometry foundation models such as MVSNet based method [2], VGGT, or MapAnything [1] to obtain denser and more accurate 3D points. In general, these geometry-oriented models are capable of producing high-quality 3D structure.
> >
> >      **[A4]** Thank you for your suggestion. It should be noted that I do not believe VGGT and our approach are contradictory. I consider VGGT's work to be analogous to the process of predicting denser point clouds in SfM or MVS, which fundamentally remains based on initial point cloud reconstruction under visible views. The key distinction of our approach lies in addressing the limitation that these methods lack the human imagination for imagery generation as scene completion. The proposed point cloud completion network can effectively fill in partial point clouds from unobserved viewpoints, as demonstrated by its capability to generate results from just 1/4 of the initial points.
> >
> > - **[Q5]** This paper did not make comparison with really SOTA such as Difix3D+ and Genfusion.
> >
> >     **[A5]** We proposed the comparsion about more challenging datasets (Mipnerf 360) with the sota methods, such as FSGS, DropGaussian and the **generative-based** methods: Difix3D+, GenFusion, GSFixer. And our GenCoGS achieved the consistent improvement, which demonstrated the generalization of our methods. And, we have check the results about each scene in both LLFF, DTU, and Mipnerf360, all results achieved consistent improvements than baseline.
> >
> >      | Method       | PSNR (24-view) | SSIM (24-view) | LPIPS (24-view) | PSNR (9-view) | SSIM (9-view) | LPIPS (9-view) |
> >      | ------------ | -------------- | -------------- | --------------- | ------------- | ------------- | -------------- |
> >      | E3DGS        | 22.80          | 0.708          | 0.276           | -             | -             | -              |
> >      | FSGS         | 23.70          | 0.745          | 0.230           | 17.94         | 0.492         | 0.468          |
> >      | CoR-GS       | 23.39          | 0.727          | 0.271           | -             | -             | -              |
> >      | DropGaussian | 24.13          | 0.762          | 0.225           | -             | -             | -              |
> >      | ReconFusion  | -              | -              | -               | 18.19         | 0.432         | 0.511          |
> >      | 3DGS         | -              | -              | -               | 16.79         | 0.447         | 0.446          |
> >      | Difix3D+     | -              | -              | -               | 17.54         | 0.452         | 0.391          |
> >      | GenFusion    | -              | -              | -               | 18.29         | 0.489         | 0.440          |
> >      | GSFixer      | -              | -              | -               | 18.63         | 0.481         | 0.420          |
> >      | **GenCoGS**  | **24.37**      | **0.757**      | **0.218**       | **20.1473**   | **0.6109**    | **0.3230**     |

---

### Official Review · Reviewer_YRYJ · 2025-10-29

**Soundness:** 2
**Presentation:** 2
**Contribution:** 2
**Rating:** 4
**Confidence:** 3

**Summary:**

GenCoGS proposes to improve few-shot novel-view synthesis with 3D Gaussian Splatting with a two-step procedure. First, generative point-cloud completion: a generate-and-filter complementary-point cloud to densify the SfM point cloud. Second, generative pseudo-view generation: sampling pseudo camera poses via a perturbed camera trajectory, producing conditional completions with an image-to-video diffusion model. The authors report consistent gains across LLFF/DTU/Shiny and present ablation results showing that each component helps.

**Strengths:**

1. Clear motivation. The paper tackles a real failure mode for few-shot 3DGS methods - poor scene completion and floating artifacts - and proposes a strategy to address it.
2. Two-stage design. Separating initialization (complementary points) from optimization (pseudo-view completions) is sensible and interpretable.
3. Practical ablations. The paper includes a comprehensive ablation suite. Supplementary material is particularly strong.
4. Quantitative gains. The method shows consistent rendering quality improvement over all the datasets

**Weaknesses:**

1. Novelty. The paper currently presents GenCoGS as a "generative completion" approach, but most subcomponents (point completion, filtering, pseudo-view generation, consistency loss) resemble adaptations of existing ideas. Without clearer theoretical or algorithmic justification, the method may seem as an engineering combination of known blocks.

2. Complementary Point Generator (CPG). The paper doesn’t clearly specify whether the CPG network is trained offline (on a dataset of partial -> complete point clouds) or per scene, nor the loss functions used. This directly affects reproducibility and interpretability of the paper.

3. Complimentary Point Filtering (CPF). CPF uses a global distance threshold to remove outliers in generated points. This is brittle and scene-dependent; the ablation shows sensitivity. It does not seem to be robust enough to be generalizable.

4. Evaluation. Metrics are reported from single runs without standard deviation. The paper also claims hallucination attenuation but provides no quantitative measure of it.

5. Practicality. Training takes 3x longer and uses 2.5x more memory than vanilla 3DGS, yet the paper underplays this limitation. It is important to know whether GenCoGS is practical.

For every point, see the Questions section for suggestions.

**Questions:**

For the points mentioned in the Weaknesses:

1. Novelty. Explicitly state the unique contribution in one paragraph

2. CPG. Elaborate on CPG training details such as:
2.a. Training details of CPG
2.b  Clarify whether CPG is pretrained once and then frozen, or jointly optimized per scene
2.c If trained per-scene, provide the time overhead and discuss how this scales
2.d Explain the intuition behind all the methods used to generate the complimentary points: DGCNN, PE, kNN, Dynamic Query Mechanism, FoldingNet

3. CPF. CPF seems like a fine-tuning trick with a sensitive threshold:
3a. Provide ablations on sensitivity of the delta parameter over other datasets and different amount of views
3b. Provide the statistics on what % of the newly added points are filtered out

4. Evaluation. The whole setup with relatively small gains seems "seed dendent". Therefore
4a. Provide mean and std for all the metrics over 5 runs at least on LLFF dataset
4b. Can the authors somehow quantify the hallucination and its attenuation that was achieved using your approach?

5. Practicality. Can you provide compute-runtime/quality tradeoff? For example, on the X axis number of iterations/number of generated views and on the Y PSNR?

---

> ### Author Response · Authors · 2025-11-25
> **Response to Reviewer YRYJ - Part(1/2)**
>
> We sincerely thank the reviewer for the careful evaluation, insightful observations, and constructive questions. We appreciate the recognition of our motivation, two-stage design, and experimental thoroughness, and we address all concerns regarding novelty, CPG/CPF details, evaluation stability, and practicality in the responses below.
>
> - **[Q1]** Novelty. Explicitly state the unique contribution in one paragraph
>
>    **[A1]** The core novelty of **GenCoGS** lies in its unified framework that **generative completion** with focus on **initializing and optimizing** scene representation to enhance 3DGS from sparse or few-shot input views in line with the **mechanism of human imagination**.
>
>     1), **GCGI (point-cloud completion)**, generates complementary 3D points to fill unobserved regions, providing a more complete initial coverage for 3D Gaussian Splatting with the **Complementary Point Filtering**, which can address this **points hallucination** while maintaining the scene's structural details.
>
>     2), **Generative Pseudo-View Completion-Based Gaussian Optimization (GCGO, §3.2)**, leverages **generatively produced pseudo-views** to supervise Gaussian parameter optimization, while explicitly addressing potential multi-view inconsistencies and hallucinations via two mechanisms: **Perturbed Camera Trajectory**, which augments the pseudo-view supervision with small camera perturbations to enforce robust multi-view consistency, and **Generative Consistency Loss**, which constrains the optimization to remain faithful to the underlying scene geometry and appearance across views. By integrating these strategies, GenCoGS **directly injects generative priors into both the initialization and optimization of 3D representations**, ensuring high-fidelity, multi-view-consistent reconstructions and enabling robust novel-view synthesis in sparse-view and unbounded scenarios, distinguishing it from prior works that only enhance 2D renderings or rely on single-stage generative reconstruction.
>
> - **[Q2]** CPG. Elaborate on CPG training details such as: 2.a. Training details of CPG 2.b Clarify whether CPG is pretrained once and then frozen, or jointly optimized per scene 2.c If trained per-scene, provide the time overhead and discuss how this scales 2.d Explain the intuition behind all the methods used to generate the complimentary points: DGCNN, PE, kNN, Dynamic Query Mechanism, FoldingNet
>
>     **[A2]** We first answer the question about training details, The CPG module is trained on the ShapeNet dataset in a standard supervised manner, It is optimized using Adam with a learning rate of 1 e- 4 1e-4, batch size of 24, and Chamfer Distance as the reconstruction loss. The module receives sparse point clouds (simulated from our training distribution) and predicts their completed counterparts. To ensure stable convergence, we additionally apply coordinate normalization (centering and scaling), following common practices in point cloud completion. Aoth of CPG is frozen, and no need for optimized per scene.
>
> - **[Q3]** CPF seems like a fine-tuning trick with a sensitive threshold: 3a. Provide ablations on sensitivity of the delta parameter over other datasets and different amount of views 3b. Provide the statistics on what % of the newly added points are filtered out
>
>     **[A3]** We provided the ablation about $\delta_{1}$ in DTU and the filter percent $\%$ in LLFF in below Tables. As we mentioned in our paper, although there is some fluctuation in the results depending on different values of $\delta_1$, the consistent improvement in LPIPS and PSNR metrics relative to the method without CPF is maintained. Furthermore, our choice of $\delta_1 = 1$ is also consistent with the initial condition d2=minj∥pi−pj∥2 in 3DGS, which aligns with a certain degree of interpretability.
>
>      | \$\delta\_{1}\$ | PSNR (DTU) | LPIPS | SSIM | Filter (%) |
>      |--------------|------------|-------|-------|------------|
>      | 0.5          | 22.95      | 0.093 | 0.904 | 16.7       |
>      | 1.0          | 23.11      | 0.082 | 0.910 | 11.1       |
>      | 2.0          | 22.86      | 0.091 | 0.901 | 5.1        |
>
>      | \$\delta_{1}\$         | 0.5   | 1.0   | 2.0   |
>      | -------------------- | ----- | ----- | ----- |
>      | PSNR (LLFF)          | 22.09 | 22.13 | 22.03 |
>      | Filter percent($\%$) | 15.3  | 11.9  | 3.4   |

---

> ### Author Response · Authors · 2025-11-25
> **Response to Reviewer YRYJ - Part(2/2)**
>
> - **[Q4]** Evaluation. The whole setup with relatively small gains seems "seed dendent". Therefore 4a. Provide mean and std for all the metrics over 5 runs at least on LLFF dataset 4b. Can the authors somehow quantify the hallucination and its attenuation that was achieved using your approach?
>
>      **[A4]** 4a. Thank you for your suggestion. We have provided **over 3 additional runs on the LLFF datasets**. The results are as follows: the results demonstrated that the proposed method exhibits high stability, with a **mean of 22.11 and a variance of only 0.05**.
>
>      | LLFF                     | PSNR                 | LPIPS                  | SSIM                    | AVGE                   |
>      | ------------------------ | --------------------- | --------------------- | ----------------------- | ---------------------- |
>      | **Test1**                | 22.08                 | 0.166                 | 0.764                   | 0.085                  |
>      | **Test2**                | 22.18                 | 0.168                 | 0.764                   | 0.084                  |
>      | **Test3**                | 22.06                 | 0.162                 | 0.767                   | 0.083                  |
>      | **Test4** (in our paper) | 22.13                 | 0.164                 | 0.762                   | 0.084                  |
>      | **Mean ± Std**           | **22.1125 ± 0.05377** | **0.1650 ± 0.002582** | **0.76425 ± 0.0020616** | **0.0840 ± 0.0008165** |
>
>      4b. We conducted comparision using FID as a metric for evaluating generated views, with the results shown below. The **bolded** parameters represent our selected hyperparameters. As observed, our chosen hyperparameters yield superior generation quality compared to other options, effectively eliminating potential hallucinations. Additionally, our GenCoGS achieves greater consistency in generated images compared to those produced by Difix3D+ under identical trajectories.
>
>
>      |                              | PSNR\$\uparrow\$ | FID\$\downarrow\$ |
>      | ---------------------------- | -------------- | --------------- |
>      | Difix3D+                     | 21.96          | 66.70           |
>      | GenCoGS（A=3.0, f = 1.0）    | 21.83          | 68.92           |
>      | GenCoGS（A=2.0, f = 2.0）    | 22.06          | 62.34           |
>      | **GenCoGS**（A=2.0, f = 1.0) | 22.13          | 58.79           |
>
> - **[Q5]** Practicality. Can you provide compute-runtime/quality tradeoff? For example, on the X axis number of iterations/number of generated views and on the Y PSNR?
>
>      **[A5]** We provided the computation/runtime/quality trade-off ablation about the pseudo-views number, you can see the most time costing is from the diffusion model, with the experiments in **R1[A5]**, you can use the one-step diffusion model as Difix3D for better effeciency.
>
>      | Views_Number  | Training_Time | PSNR  |
>      | ------------- | ------------- | ----- |
>      | 12            | 24min         | 21.86 |
>      | 24            | 32min         | 22.01 |
>      | 36 **(Ours)** | 40min         | 22.13 |
>      | 60            | 90min         | 22.24 |

---

### Official Review · Reviewer_5er1 · 2025-10-31

**Soundness:** 3
**Presentation:** 3
**Contribution:** 3
**Rating:** 4
**Confidence:** 4

**Summary:**

This paper proposes GenCoGS, a generative completion-based framework that enhances few-shot novel view synthesis (NVS) using 3D Gaussian Splatting (3DGS).

(1) Generative Point Cloud Completion-based Gaussian Initialization (GCGI) that refines Gaussian initialization through complementary point generation and filtering

(2)Generative Pseudo View Completion-based Gaussian Optimization (GCGO), employing diffusion-based pseudo-view synthesis and a generative consistency loss.

Experiments on LLFF, DTU, and Shiny datasets show consistent improvements in PSNR, SSIM, and LPIPS over baselines like FSGS and BinoGS.

**Strengths:**

1.	Combines complementary strategies (GCGI + GCGO) into a coherent and unified 3DGS framework, improving both initialization and optimization phases.

2.	Conducts comprehensive experiments across multiple datasets and shot settings, showing clear and consistent gains over strong baselines.

3.	Demonstrates reduced artifacts and more complete scene reconstruction in unobserved regions, indicating effective integration of generative priors for scene completion.

**Weaknesses:**

1.	Limited conceptual novelty: The method primarily extends prior diffusion-augmented NVS ideas (e.g., ViewCrafter, ReconFusion) to point cloud based 3DGS, without introducing a substantially new theoretical concept.

2.	Weak theoretical grounding: The paper lacks formal analysis or intuition on how generative completion improves geometric consistency or the optimization dynamics of Gaussian Splatting.

3.	External priors: The diffusion-based pseudo-view completion uses a pre-trained I2V model as a black box, but the paper does not analyze its domain sensitivity, generalization, or contribution through ablation.

4.	Presentation clarity: The paper is technically dense, and mathematical sections could be simplified or supplemented with a clearer pipeline overview or pseudocode for better readability.

**Questions:**

1.	How does GenCoGS compare computationally (training time and GPU memory) with FSGS and BinoGS?

2.	Could the Complementary Point Filtering (CPF) module inadvertently remove valid geometry in textured or noisy regions?

3.	Could the authors test on more challenging or real-world datasets, and include failure case analysis to assess generalization limits?

---

> ### Author Response · Authors · 2025-11-25
> **Response to Reviewer 5er1 - Part(1/2)**
>
> We thank the reviewer for the thoughtful evaluation and the constructive comments. We appreciate the recognition of our framework’s strengths and the detailed feedback regarding clarity and analysis. We address all concerns point-by-point below.
>
> - **[Q1]** Limited novelty and weak theoretical grounding about the optimization of Gaussian Splatting：
>
>   **[A1]** As for limited novelty, we are the first work to generatvie complete the both 3DGS initialization and optimization with the mechanism of human imagination. And we proposed the GCGI, and GCGO (an pseudo view completion module) with the novel **Perturbed Camera Trajectory** and **Generative Consistency Loss** to suppress the hallucination.
>
> - **[Q2]** External priors: The diffusion-based pseudo-view completion uses a pre-trained I2V model as a black box, but the paper does not analyze its domain sensitivity, generalization, or contribution through ablation.
>
>   **[A2]** As we mentioned in the manuscript, both the completion point cloud model in CGCI and the I2V model in CGCO are pre-trained on large-scale open-source datasets, thus exhibiting a certain degree of generalization. Meanwhile, This paper employs CPF and PERTURBED CAMERA TRAJECTORY and GENERATIVE CONSISTENCY LOSS—which can be regarded as part of our main contributions—to mitigate the impact of generalization issues caused by potential hallucinations in black-box models. Furthermore, we conducted additional experiments on datasets such as MIPNerf360 in **[Q3]**, with consistent results validating the effectiveness of the proposed method.
>
> - **[Q3]** How does GenCoGS compare computationally (training time and GPU memory) with FSGS and BinoGS?
>
>   **[A3]** We discuss the comparison of computational complexity and the discussion between the proposed GenCoGS and existing state-of-the-art methods in Table 10 of the Appendix, as shown in the Table below.
>
>     | Method                       | PSNR ↑    | SSIM ↑    | LPIPS ↓   | Training Time ↓ | Training Memory ↓ |
>     | ---------------------------- | --------- | --------- | --------- | --------------- | ----------------- |
>     | 3DGS                         | 15.52     | 0.405     | 0.408     | **13.0 min**    | **1.6 GB**        |
>     | DNGaussian (Li et al., 2024) | 19.12     | 0.591     | 0.294     | 23.5 min        | 2.0 GB            |
>     | FSGS                         | 20.31     | 0.652     | 0.288     | 28.0 min        | 2.4 GB            |
>     | BinoGS (Han et al., 2024)    | 21.44     | 0.751     | 0.168     | 30.0 min        | 3.0 GB            |
>     | **Ours**                     | **22.13** | **0.762** | **0.164** | 40.0 min        | 4.0 GB            |
>
> - **[Q4]** Could the Complementary Point Filtering (CPF) module inadvertently remove valid geometry in textured or noisy regions?  $\surd$
>
>   **[A4]** Thank you for your suggestion. As described in Section 3.1.2, our proposed CPF method is based on KD-Tree, preserving geometric consistency as much as possible while performing filter operations that adaptively utilize the **“average scale”** of high-quality SfM initial points. These two aspects inherently contribute to effective structural consistency preservation, a fact further validated by subsequent experimental results.Of course, the proposed method may incur some loss in certain textures/colors. However, crucial color information will be optimized during the subsequent 3DGS optimization process.

---

> > ### Author Response · Authors · 2025-11-25
> > **Response to Reviewer 5er1 - Part(2/2)**
> >
> > - **[Q5]** Could the authors test on more challenging or real-world datasets, and include failure case analysis to assess generalization limits?
> >
> >    **[A5]** We proposed the comparsion about more challenging datasets (Mipnerf 360) with the sota methods, such as FSGS, DropGaussian and the **generative-based** methods: Difix3D+, GenFusion, GSFixer. And our GenCoGS achieved the consistent improvement, which demonstrated the generalization of our methods. And, we have check the results about each scene in both LLFF, DTU, and Mipnerf360, all results achieved consistent improvements than baseline.
> >
> >      | Method       | PSNR (24-view) | SSIM (24-view) | LPIPS (24-view) | PSNR (9-view) | SSIM (9-view) | LPIPS (9-view) |
> >      | ------------ | -------------- | -------------- | --------------- | ------------- | ------------- | -------------- |
> >      | E3DGS        | 22.80          | 0.708          | 0.276           | -             | -             | -              |
> >      | FSGS         | 23.70          | 0.745          | 0.230           | 17.94         | 0.492         | 0.468          |
> >      | CoR-GS       | 23.39          | 0.727          | 0.271           | -             | -             | -              |
> >      | DropGaussian | 24.13          | 0.762          | 0.225           | -             | -             | -              |
> >      | ReconFusion  | -              | -              | -               | 18.19         | 0.432         | 0.511          |
> >      | 3DGS         | -              | -              | -               | 16.79         | 0.447         | 0.446          |
> >      | Difix3D+     | -              | -              | -               | 17.54         | 0.452         | 0.391          |
> >      | GenFusion    | -              | -              | -               | 18.29         | 0.489         | 0.440          |
> >      | GSFixer      | -              | -              | -               | 18.63         | 0.481         | 0.420          |
> >      | **GenCoGS**  | **24.37**      | **0.757**      | **0.218**       | **20.1473**   | **0.6109**    | **0.3230**     |

---

### Official Review · Reviewer_tyFj · 2025-10-31

**Soundness:** 3
**Presentation:** 3
**Contribution:** 3
**Rating:** 6
**Confidence:** 5

**Summary:**

This work proposes a generative-model-based initialization and optimization framework for few-shot novel view synthesis. Specifically, a set-to-set generation module is used to complete the SfM-reconstructed point cloud, and an image-to-video diffusion model is employed to produce pseudo-views for supervision. Extensive experiments demonstrate the effectiveness of the proposed approach.

**Strengths:**

- The paper is clearly organized, making it easy to follow.
- The proposed initialization strategy is interesting, as it leverages generative models to complete point clouds.
- Both qualitative and quantitative results demonstrate the effectiveness of the proposed method.

**Weaknesses:**

- Several important implementation details of GCGI are missing: (1) the computation of the positional embedding in Eq. 1. (2) the initialization strategy and the number of dynamic queries Q. (3) whether DGCNN, $M_E$, $M_D$, and Foldingnet are pretrained. (4) the structure details of $M_E$ and $M_D$. (5) in the top-left of Fig. 2, whether “encoder-decoder” refers to $M_E$/$M_D$ or FolderNet.
- The same mathematical symbol is overloaded to denote multiple concepts, which may cause confusion. For example, $m$ in Line 180 and Line 321, $C$ in Line 164 and Line 284.

**Questions:**

- This work states that GCGI 'produces and filters complementary points toward a complete point cloud with refined structural and appearance information for Gaussian initialization'. However, Sec. 3.1 primarily describes structural point cloud completion for initialization. Could the authors clarify how the appearance information is refined within this module?
- GIGC aims to achieve set-to-set generation. Have the authors considered an alternative pipeline in which image-to-video (I2V) generation is first applied, followed by structure-from-motion (SfM) on both the real and generated views to obtain a more complete point cloud? Besides, my concern is that the set-to-set generation models is a bit out-of-date, does it perform good than the recent I2V generation models?
- Could the authors clarify the differences between the proposed GCGO and the similar techniques used in 3DGS-Enhancer [A] and DIFIX3D+ [B]?
- In Eq. 18, $L_{reg}$ applies an L1 loss on pseudo views, which also filters the noises. Why does the $L_{img}$, used at the same stage, also adopt an L1​ loss that does not filter noise? These two losses may be redundant—could the authors clarify the motivation behind them?

[A] Liu, Xi, Chaoyi Zhou, and Siyu Huang. "3dgs-enhancer: Enhancing unbounded 3d gaussian splatting with view-consistent 2d diffusion priors." NeurIPS 2024.

[B] Wu, Jay Zhangjie, et al. "Difix3d+: Improving 3d reconstructions with single-step diffusion models."CVPR 2025.

---

> ### Author Response · Authors · 2025-11-25
> **Response to Reviewer tyFj - Part(1/2)**
>
> Thank you very much for your thoughtful and constructive review. We greatly appreciate the time and effort you spent analyzing our work and providing detailed feedback. Below, we address your points and clarify the issues you raised.
>
> - **[Q1]** Several important implementation details of GCGI are missing: (1) the computation of the positional embedding in Eq. 1. (2) the initialization strategy and the number of dynamic queries Q. (3) whether DGCNN, $M_E$,$M_D$,  and Foldingnet are pretrained. (4) the structure details of $M_E$ and $M_D$. (5) in the top-left of Fig. 2, whether “encoder-decoder” refers to $M_E$/$M_D$ or FolderNet.
>
>
>   **[A1]** We apologize for the lack of some implementation details here. Once the paper is successfully accepted, we will open-source all code to facilitate reproducibility. For now, we supplement the relevant descriptions as follows:
>
>   As mentioned in L161, we drew inspiration from previous methods. For details:
>   1). For each point proxy, we employed an MLP as the position embedding in Eq. 1.
>
>   2). In our experimental setup, we set the number of queries to 256 as a suitable hyperparameter. Initialization followed the random initialization approach used in works like DETR.
>
>   3). All network parameters mentioned in the CPG section of this work, including DGCNN, $M_E$, $M_D$, and Foldingnet, were pretrained on the SHAPENET55 dataset. Their effectiveness was subsequently tested on datasets including LLFF, DTU, and Mipnerf360, demonstrating strong generalization capabilities.
>
>   4). We set the depth of the encoder and decoder in our transformer to 6 and 8, respectively, with each transformer layer containing 8 blocks.
>
>   5). Encoder-decoder refers to $M_E$, $M_D$. For brevity, we omit certain non-essential details, including DGCNN and Foldingnet, to highlight our core contributions such as CPF.
>
> - **[Q2]** The same mathematical symbol is overloaded to denote multiple concepts, which may cause confusion. For example, $m$ in Line 180 and Line 321, $C$ in Line 164 and Line 284.
>
>   **[A2]** Thank you for your suggestion. Due to the presence of numerous formulas in this paper to enhance intuitive understanding. Therefore, some mathematical symbols are confused in sections 3.1 and 3.2 of the method section. To facilitate subsequent reading, we have modified the original color $C$ into $RGB$ and the first phase iteration $m$ into $s$.
>
> - **[Q3]** This work states that GCGI produces and filters complementary points toward a complete point cloud with refined structural and appearance information for Gaussian initialization'. However, Sec. 3.1 primarily describes structural point cloud completion for initialization. Could the authors clarify how the appearance information is refined within this module?
>
>   **[A3]** Thank you for pointing out the potential confusion. The goal of GCGI is not to directly reconstruct textures or other appearance attributes, but to ensure that **newly completed points acquire reasonable appearance attributes**.
>
>   Specific approach (L229–231):
>
>   - After filtering, we **copy and assign** the **color/rotation/scale** attributes from reliable points within the neighborhood to the **reconstructed points**.
>   - This guarantees appearance information remains stable and avoids introducing noise.
>
>   Thus, GCGI functions as **structural reconstruction + reasonable attribute propagation**, achieving higher-quality initialization.
>
> - **[Q4]** GIGC aims to achieve set-to-set generation. Have the authors considered an alternative pipeline in which image-to-video (I2V) generation is first applied, followed by structure-from-motion (SfM) on both the real and generated views to obtain a more complete point cloud? Besides, my concern is that the set-to-set generation models is a bit out-of-date, does it perform good than the recent I2V generation models?
>
>   **[A4]** We avoid the approach about SfM pipeline you mentioned (reality + generation) because if the generated video or view which relying with the robustness of the I2V model, and may exhibit misalignment (geometric inconsistencies in the render view), blurriness, or unrealistic appearance, these pseudo-views may mislead the optimization process, leading to erroneous point clouds.
>   On the other head, such a two-stage SfM strategy incurs significant computational costs.
>
>   However, we believe it could be an interesting direction in the future as the accuracy of I2V generation improves.

---

> > ### Author Response · Authors · 2025-11-25
> > **Response to Reviewer tyFj - Part(2/2)**
> >
> > - **[Q5]** Could the authors clarify the differences between the proposed GCGO and the similar techniques used in 3DGS-Enhancer [A] and DIFIX3D+ [B]?
> >
> >   **[A5]** Thank you for your suggestions. The core contribution of the method proposed by GCGO lies in its optimization strategy of training using a diffusion model as a pseudo view completion, combined with **Perturbed Camera Trajectory** and **Generative Consistency Loss**. Since the diffusion model is replaceable, we opted to substitute it with difix3d+ for testing. The consistent results conclusively validate the effectiveness of GenCoGS's contribution:
> >
> >   | Method                  | PSNR  | SSIM  | LPIPS |
> >   | ----------------------- | ----- | ----- | ----- |
> >   | Baseline + GCGI         | 21.45 | 0.757 | 0.168 |
> >   | Baseline + GCGI + Difix | 21.96 | 0.769 | 0.164 |
> >   | Baseline + GCGI + GCGO  | 22.13 | 0.762 | 0.164 |
> >
> > - **[Q6]** In Eq. 18, $L_{reg}$ applies an L1 loss on pseudo views, which also filters the noises. Why does the $L_{img}$ , used at the same stage, also adopt an L1 loss that does not filter noise? These two losses may be redundant—could the authors clarify the motivation behind them?
> >
> >   **[A6]** This is indeed an omitted detail not explicitly stated in our paper, which caused your confusing. These two components are not redundant, as the former (L1 loss) aims to constrain the overall rendering of the refined pseudo view to enhance reconstruction of the 3DGS, while the latter focuses on specific regions to suppress hallucinations. Setting $\alpha$ = 10.0 to impose strong constraints aligns with our design rationale.
> >
> >
> >
> > [A] Liu, Xi, Chaoyi Zhou, and Siyu Huang. "3dgs-enhancer: Enhancing unbounded 3d gaussian splatting with view-consistent 2d diffusion priors." NeurIPS 2024.
> >
> > [B] Wu, Jay Zhangjie, et al. "Difix3d+: Improving 3d reconstructions with single-step diffusion models."CVPR 2025.

---

> > > ### Comment · Reviewer_tyFj · 2025-11-26
> > >
> > > I have the following follow-up questions:
> > >
> > > - Regarding *A1*: ShapeNet55 is an object-level dataset. Do DGCNN, $M_E$, $M_D$, and Foldingnet—when trained on ShapeNet55—generalize well to scene-level datasets such as LLFF and MipNeRF360? Furthermore, it would be better to label the encoder and decoder as $M_E$ and $M_D$ in Fig. 2 for clarity.
> > > - Regarding *A4*: this work also depends on the robustness of the encoder $M_E$ and decoder $M_D$ for the set-to-set generation. Given this dependency, it is difficult to conclude that set-to-set generation is superior to methods based on image generation followed by SfM.

---

> > > > ### Author Response · Authors · 2025-11-29
> > > > **Response to Reviewer tyFj**
> > > >
> > > > Thank you very much for your time and constructive feedback.
> > > >
> > > > **[Response 1]** We appreciate your suggestion regarding the pipeline in Figure 2. In the revised version, we have added the encoder and decoder notations, $\mathcal{M}_E$ and $\mathcal{M}_D$, to improve clarity.
> > > >
> > > > Regarding the generalization ability of our pre-trained CPG model from ShapeNet55 to LLFF/MipNeRF360 scenes, We conducted an in-depth analysis as follows:
> > > >
> > > > Our set-to-set completion module (CPG) is designed to perform **local geometric reasoning and completion**, rather than object- or scene-centric reconstruction. Object-level datasets can provide superior capture of local geometric details; thus, whether task is an object-level DTU or a scene-level LLFF/MipNeRF360 3DGS reconstruction does not substantially affect the points completion and achieve the consistent improvements. Our experiments also demonstrated this results.
> > > >
> > > > Furthermoe, the DGCNN backbone and Transformer layers operate on neighborhoods defined by relative coordinates and kNN, giving the encoder strict **translation invariance** and **permutation invariance**. As a result, the model primarily learns **cross-category geometric primitives**—such as smooth surfaces, edge contours, and slender structures—that appear in both ShapeNet objects and SfM-derived scene fragments. The network does not rely on global object semantics, which explains its ability to generalize across datasets.
> > > >
> > > > However, we acknowledge that training only on object-level data overlooks specific characteristics of real-world scenes, including clutter, occlusions, and complex multi-object interactions that do not exist in ShapeNet55. These distribution differences may lead the generative decoder to occasionally produce redundant or noisy points.
> > > >
> > > > To mitigate this, the introduced **Complementary Point Filter (CPF)** based on kd-tree based local anchor distances, which effectively removes outliers before the Gaussian initialization stage.
> > > >
> > > > As we mentioned, exploring the **scene-level pre-training** is a valuable direction for future research. To construct a large-scale training datasets that transitions from object-level to scene-level point clouds in a coarse-to-fine manner. Strengthening the completion module with such datasets could further enhance robustness in complex real-world environments.
> > > >
> > > > We plan to conduct training on large-scale scene-level datasets in subsequent work and will include additional experimental results in our open-source repository.
> > > >
> > > >
> > > >
> > > > **[Response to 2]** We apologize for the confusion caused by our previous explanation. Our intention was to clarify two key limitations that prevent the I2V-SfM pipeline from achieving superior than set-to-set point completion.
> > > >
> > > > 1. **I2V-generated pseudo views inevitably contain hallucinations.**
> > > >    These hallucinated structures are often subtle but *fatal* for COLMAP-based SfM, which depends on precise cross-view feature consistency as key points matching. Even minor geometric inconsistencies break feature matching and lead to unreliable triangulation.
> > > > 2. **I2V outputs images with a fixed resolution, which must be resized to match our training pipeline.**
> > > >     While such resizing is acceptable for supervision losses, it significantly distorts feature statistics used in SfM. This further degrades keypoint matching robustness across views.
> > > >
> > > > Due to the above two issues, our experiments show that SfM reconstruction using **(real + generated)** views suffers from a *drastic reduction in reconstructed points cloud*. In several scenes, the reconstructed point cloud contains only *limited than 100* of points, making the initialization highly unstable and resulting in poor NVS performance.
> > > >
> > > > The quantitative results of using the I2V → SfM pipeline are shown below:
> > > >
> > > > | LLFF （3views）        | PSNR      | SSIM      | LPIPS     | AVGE  |
> > > > | ---------------------- | --------- | --------- | --------- | ----- |
> > > > | I2V SfM                | 21.53     | 0.738     | 0.194     | 0.093 |
> > > > | **Set-to-Set（Ours）** | **22.13** | **0.762** | **0.164** | 0.084 |

---

### Author Response · Authors · 2025-11-25
**Response to All Reviewers**

We thank all reviewers for their valuable comments, which help us improve and clarify our work. If the reviewers would like to see additional results or analyses, we would be happy to provide them and are always open to further discussion.

1. Compare with sota methods such as Difix3D+， GenFusion，GSFixer on Mipnerf 360 results:

   | Method       | PSNR (24-view) | SSIM (24-view) | LPIPS (24-view) | PSNR (9-view) | SSIM (9-view) | LPIPS (9-view) |
   | ------------ | -------------- | -------------- | --------------- | ------------- | ------------- | -------------- |
   | E3DGS        | 22.80          | 0.708          | 0.276           | -             | -             | -              |
   | FSGS         | 23.70          | 0.745          | 0.230           | 17.94         | 0.492         | 0.468          |
   | CoR-GS       | 23.39          | 0.727          | 0.271           | -             | -             | -              |
   | DropGaussian | 24.13          | 0.762          | 0.225           | -             | -             | -              |
   | ReconFusion  | -              | -              | -               | 18.19         | 0.432         | 0.511          |
   | 3DGS         | -              | -              | -               | 16.79         | 0.447         | 0.446          |
   | Difix3D+     | -              | -              | -               | 17.54         | 0.452         | 0.391          |
   | GenFusion    | -              | -              | -               | 18.29         | 0.489         | 0.440          |
   | GSFixer      | -              | -              | -               | 18.63         | 0.481         | 0.420          |
   | **GenCoGS**  | **24.37**      | **0.757**      | **0.218**       | **20.1473**   | **0.6109**    | **0.3230**     |



2. Novelty:

   Motivation:  Inspired by the **mechanism of human imagination**, we propose a unified few-shot NVS method based on **generative completion** with focus on **initializing and optimizing** scene representation.


   The core novelty of **GenCoGS** lies in its unified framework that **generative completion** with focus on **initializing and optimizing** scene representation to enhance 3DGS from sparse or few-shot input views in line with the **mechanism of human imagination**.

   1), **GCGI (point-cloud completion)**, generates complementary 3D points to fill unobserved regions, providing a more complete initial coverage for 3D Gaussian Splatting with the **Complementary Point Filtering**, which can address this **points hallucination** while maintaining the scene's structural details.

   2), **Generative Pseudo-View Completion-Based Gaussian Optimization (GCGO, §3.2)**, leverages **generatively produced pseudo-views** to supervise Gaussian parameter optimization, while explicitly addressing potential multi-view inconsistencies and hallucinations via two mechanisms: **Perturbed Camera Trajectory**, which augments the pseudo-view supervision with small camera perturbations to enforce robust multi-view consistency, and **Generative Consistency Loss**, which constrains the optimization to remain faithful to the underlying scene geometry and appearance across views. By integrating these strategies, GenCoGS **directly injects generative priors into both the initialization and optimization of 3D representations**, ensuring high-fidelity, multi-view-consistent reconstructions and enabling robust novel-view synthesis in sparse-view and unbounded scenarios, distinguishing it from prior works that only enhance 2D renderings or rely on single-stage generative reconstruction.

---

### Note · Authors · 2026-03-04

I have read and agree with the venue's withdrawal policy on behalf of myself and my co-authors.

---

### Meta-Review · Area_Chair_a5U1 · 2026-01-06

**Summary:**

Limited novelty and weak theoretical grounding (5er1, YRYJ): The method largely combines existing ideas without a clear new conceptual or theoretical contribution.

Missing or unclear implementation details affecting reproducibility (tyFj, YRYJ): Important components (e.g., embeddings, query setup, training details, network structures) are insufficiently specified.

Ambiguity around the Complementary Point Generator and Filtering (YRYJ, XGYu): Training setup, robustness, and generalization of CPG/CPF are unclear or brittle.

External priors not well-analyzed (5er1, XGYu): The reliance on diffusion/I2V priors lacks ablation and discussion of domain sensitivity or consistency.

Evaluation concerns (YRYJ, XGYu): Missing standard deviations, unclear baselines in ablations, and lack of comparison to stronger SOTA methods.

Practical limitations (YRYJ): The method is significantly slower and more memory-intensive than vanilla 3DGS.

**Reviewer Concerns:**

The authors have provided point-by-point responses to the reviewers’ comments, and in that sense the concerns raised during the initial review round have been addressed. However, some of the replies, particularly those pertaining to the level of novelty, are not fully convincing. This is not to say that the paper lacks novelty altogether; rather, the degree of novelty does not appear to meet the standard typically expected for this conference. Consequently, it is likely that at least some reviewers will remain unconvinced after reading the rebuttal. Overall, while the paper has merit, the contribution may not be sufficiently distinct or innovative to justify acceptance at this venue.

**Reviewer Scores:**

Reviewer tyFj: 6
Reviewer 5er1: 4
Reviewer YRYJ: 4
Reviewer XGYu: 2

Based on the authors’ rebuttal, some concerns, particularly those related to the experimental results, may have been addressed. However, major issues such as the novelty concerns are likely to remain. I do not see a strong reason for the reviewers to change their original scores.

---

### Decision · Program_Chairs · 2026-01-26

Reject